# Intelligent metasurface system for automatic tracking of moving targets and wireless communications based on computer vision

Weihan Li [1], Qian Ma[1], Che Liu[1], Yunfeng Zhang[1], Xianning Wu[2], Jiawei Wang[1], Shizhao Gao[1], Tianshuo Qiu[2], Tonghao Liu[2], Qiang Xiao[1], Jiaxuan Wei[1], Ting Ting Gu [3], Zhize Zhou[3], Fashuai Li [3], Qiang Cheng [1], Lianlin Li [4], Wenxuan Tang [1]✉ & Tie Jun Cui [1]✉

The fifth-generation (5G) wireless communication has an urgent need for target tracking. Digital programmable metasurface (DPM) may offer an intelligent and efficient solution owing to its powerful and flexible controls of electromagnetic waves and advantages of lower cost, less complexity and smaller size than the traditional antenna array. Here, we report an intelligent metasurface system to perform target tracking and wireless communications, in which computer vision integrated with a convolutional neural network (CNN) is used to automatically detect the locations of moving targets, and the dual-polarized DPM integrated with a pre-trained artificial neural network (ANN) serves to realize the smart beam tracking and wireless communications. Three groups of experiments are conducted for demonstrating the intelligent system: detection and identification of moving targets, detection of radio-frequency signals, and real-time wireless communications. The proposed method sets the stage for an integrated implementation of target identification, radio environment tracking, and wireless communications. This strategy opens up an avenue for intelligent wireless networks and self-adaptive systems.

In the fifth generation (5G) wireless communication era with a large number of devices in use, the demands for the Internet of Things (IoT) and intelligence for user-based positioning and tracking services become more urgent. Target tracking is generally based on advanced sensors, such as radar, which can detect and track targets by analyzing and processing the radar echoes of targets. However, the electromagnetic (EM) environment is complicated and changeable, and the detecting systems based on radars tend to be inefficient due to their complexity, high cost, and large volume. Therefore, more flexible hardware architecture, higher speed of information processing, and

more advanced theory of information and communication are urgently needed for satisfying the substantial demands of user positioning, tracking, and extensive applications in 5G communications.

Metamaterials have attracted great interest in the past decades owing to their remarkable EM properties[1–6]. They are mainly introduced by their subwavelength unit cells and functional arrangements to manipulate the EM behaviors, yielding many interesting phenomena and devices such as negative refraction[4] and absorber[5]. Metasurfaces are two-dimensional (2D) versions of metamaterials, which are of particular interest because of their planar profile, easy integration, and

[1]State Key Laboratory of Millimeter Waves and Institute of Electromagnetic Space, Southeast University, 210096 Nanjing, China. [2]Shaanxi Key Laboratory of Artificially-Structured Functional Materials and Devices, Air Force Engineering University, 710051 Xi'an, China. [3]State Key Laboratory of CAD & CG, Zhejiang University, 310058 Hangzhou, China. [4]State Key Laboratory of Advanced Optical Communication Systems and Networks, School of Electronics, Peking University, 100871 Beijing, China. ✉e-mail: wenxuant@seu.edu.cn; tjcui@seu.edu.cn

low loss. A series of intriguing findings of the metasurfaces have been reported recently, such as holography[7,8], ultrasensitive spectroscopy[9], nonlinear photonics[10], and quantum photonics[11], together with the exciting applications in photonic devices[12], terahertz devices and systems[13–16], and microwave engineerings like glide-symmetric devices[17], Huygens surfaces[18,19], field modulations[20,21], and nonreciprocal[22] and reconfigurable[23] components.

Digital coding and programmable metasurfaces consist of unique coding elements with discretized reflection phases (e.g., 0° and 180°) represented by digital bits (e.g., '0' and '1'), which can be used to manipulate the EM waves in a digitally discrete manner[6]. Digital elements can be tuned when active devices such as positive–intrinsic–negative (PIN) diodes and varactors are integrated, and in this way, programmable metasurfaces that possess a variety of functions under dynamical controls through different bias sequences have been developed[24]. In particular, polarization modulations[25–27], amplitude modulations[28,29], and transmission–reflection controls[30–34] of the programmable metasurfaces provide more degrees of freedom to modulate the carrier waves and lead to many intriguing findings and engineering applications, such as microwave imagers[35,36], space–time modulations[37–39], and wireless communications[40–43].

In practice, however, the majority of the programmable metasurfaces are controlled by human beings. Although several self-adaptive metasurfaces without human interventions have been developed to realize invisibility cloaks[44] and adaptively dynamic reactions[45], most of the related work is concentrated on the verification of pre-designed functions and performance. To realize an intelligent tracking system, on one hand, optimization schemes[46,47] and artificial intelligence (AI) like deep learning[48] (DL) techniques are adopted to compute the coding matrices of metasurface for complicated scattering problems[44,49–51] and real-time responses; and on the other hand, advanced intelligent wireless communication systems are also strongly demanded to capture user locations in complex EM environment, and to establish real-time channels between users. However, it is expensive and complicated to realize real-time and self-adaptive EM responses in a complex environment. Fortunately, with the fast development of computer vision technology, intuitive, reliable, informative, and cost-effective target detection and tracking become possible in many application scenarios including ship detection[52] and traffic surveillance system[53]. The major tasks of computer vision include classification, location, detection, and segmentation. Among them, the task of visual object detection[54] is to determine whether an image contains an object of interest or not. It is used to find out the objects of a specific category in a given image, and mark the positions of objects in each frame. After that, the process of object tracking[54,55] is adopted to continuously estimate the state of the object in subsequent video sequences based on the given position and size of the object in the initial frame. Moreover, AI-enabled computer vision technology[56–60] is evolving rapidly, which can solve more complicated problems and serve as an aid to intelligent communications.

In this article, we use the advantages of computer vision and flexible controls of the digital programmable metasurface (DPM) to achieve intelligent EM tracking and communications simultaneously. It is an innovative combination of the AI-based intelligent control of EM behaviors and computer-vision-based accurate classification, detection, and tracking. The technology enables real-time and accurate EM responses to meet the challenges in 5G and 6G communications. Here, we propose the concept of an intelligent tracking system using computer vision and fulfill the design with a dual-polarized DPM. Each element of the metasurface includes two loaded PIN diodes for dual polarizations. With the aid of a field programmable gate array (FPGA) that processes the coding sequences in real-time, the reflection property of each element can be independently controlled and the corresponding EM responses are thus dynamically produced.

Embedded with a pre-trained artificial neural network (ANN), DPM can respond to information at a high speed, where the information can be collected, fed back, and processed in real-time. The information includes the trained voltage sequences, surrounding background, and the moving objects to be tracked, without any human intervention. All bias voltages are automatically calculated by real-time information perception and then instantly supplied to DPM. The object tracking algorithm based on YOLOv4-tiny and the pre-trained ANN is used to imitate the real-time system in intelligent tracking. Experiments were carried out to evaluate the performance of the design, demonstrating that the DPM-based intelligent system exhibits self-adaptability to track moving targets and transmit information to them in real-time. The proposed concept will provide solutions for intelligent meta-systems and manipulations of EM waves in an unsupervised approach.

## Results

### Architecture of the intelligent scheme

The schematic of the proposed intelligent system is presented in Fig. 1, which is composed of a dual-polarized DPM and an RGB-D camera. The moving target is represented by a model car running along a certain path in time from $t_1$ to $t_2$. The images of the car are taken by an Intel RealSense Depth Camera D435i (RS-Camera) located on the metasurface at the rate of 40 FPS (frames per second), and each image is selected by the convolutional neural network (CNN) based on YOLOv4-tiny. The original image is firstly scaled to [608, 608, 3] when reasoning, and then input to the CSPDarknet53-tiny network for the feature extraction. After passing through five groups of convolution and pooling layers, the feature maps with three different dimensions are obtained, which are fused by the network and YOLO layers. We optimize the network to detect not only the target but also to collect its position and its elevation and azimuth angles in the coordinates of the RS-Camera (see Supplementary Note 1 for details).

The moving object is tracked dynamically and its position information is refreshed in real-time by the RS-Camera, with each refresh followed by a voltage control sequence that feeds the FPGA connected to DPM. The DPM is carefully designed to transmit an adaptive radiating beam toward the moving target based on its changing positions. The coordinate systems of the camera and the DPM (as they are closely located) are unified to ensure the accuracy of the position. As it is time-consuming to get all spatial angles through numerical simulation of the coding metasurfaces, an ANN is designed based on the theory of beam-steering coding metasurfaces. Through training the neural network, we can promptly obtain the coding sequences of DPM corresponding to all radiation angles that fulfill the moving space of the target. The pre-trained neural network for far-field control and the intelligent tracking algorithm based on YOLOv4-tiny have completed the main part of the whole control system. The coding sequence of the DPM is obtained through the extraction of position information using the neural network, and the refreshed coding sequence is sent to the DPM in the form of voltage through the FPGA.

### Design of DPM

In constructing the DPM, a 1-bit dual-linearly polarized element containing two PIN diodes is proposed, as shown in Fig. 2a, in which the geometric parameters are designed as follows: $a = 25$ mm, $b = 11.5$ mm, $c_1 = 6$ mm, $c_2 = 2.8$ mm, and $d = 1.5$ mm. Two PIN diodes are connected to the central patch through two metal bars along the $x$- and $y$-axes to tailor the phases in the orthogonally linear polarizations. The dielectric substrates of the element are made of commercially printed circuit boards (PCBs), in which the upper substrate is F4B with a height $h_1 = 3$ mm, dielectric constant $\varepsilon_r = 3$, and tangent loss $\tan \delta = 0.003$, and the lower substrate is F4B with a height $h_2 = 1$ mm. A metallic sheet is inserted between the two dielectric substrates. In the back layer, the positive side of the PIN diode is conducted with the backside sector structure for the radio-frequency (RF) signal isolation and DC voltage

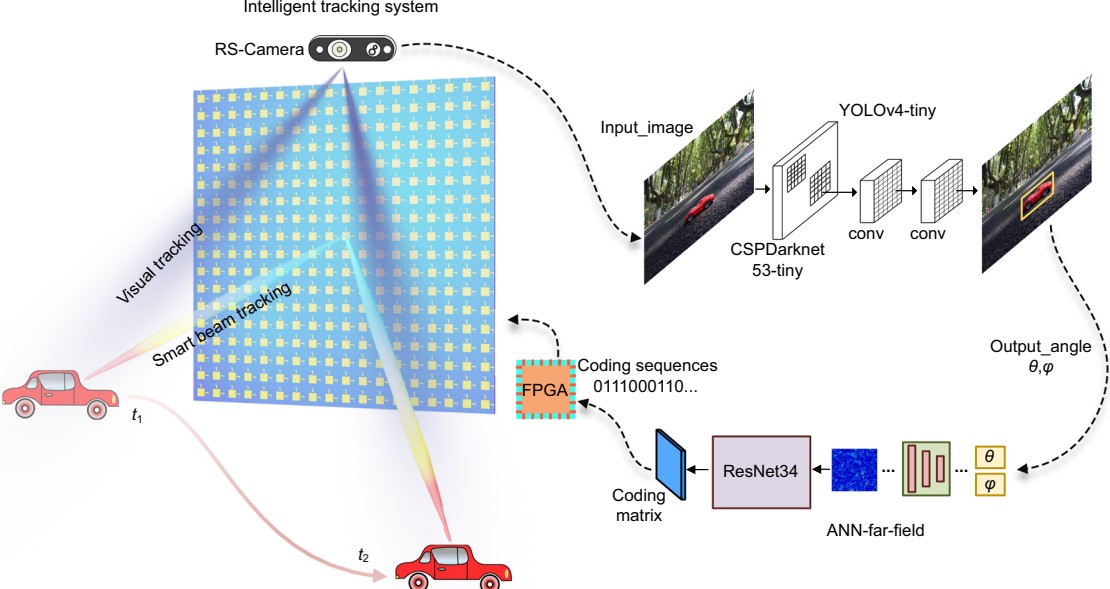

**Fig. 1 | Schematic of the intelligent target tracking system.** The RS-camera automatically detects the position of a moving target in the environment. The position information of the selected target is taken as the input of the pre-trained artificial neural network (ANN), and the coding sequence of the metasurface is output in a few milliseconds, which is sent to the dual-polarized programmable metasurface through FPGA, thus realizing smart beam tracking and wireless communications.

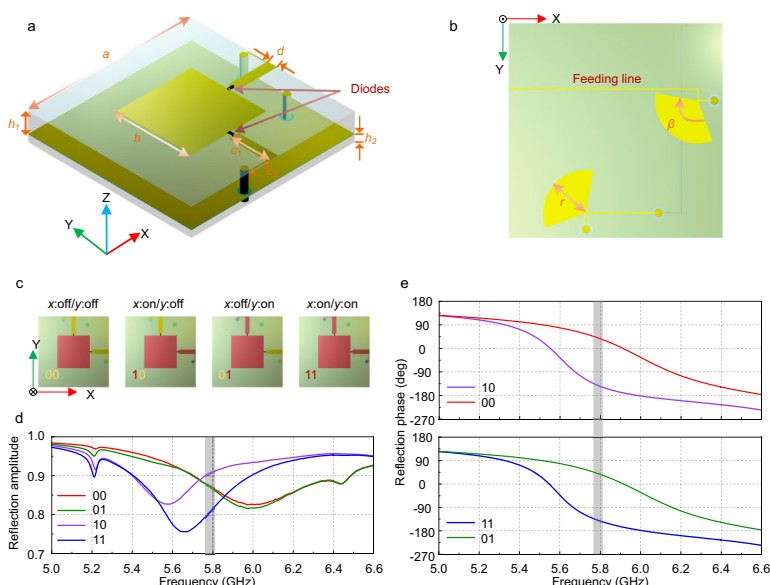

**Fig. 2 | The structure and performance of the designed DPM. a** The element structure integrated with two PIN diodes, and the geometric parameters are designed as follows: $a = 25$ mm, $b = 11.5$ mm, $c_1 = 6$ mm, $c_2 = 2.8$ mm, $h_1 = 3$ mm, $h_2 = 1$ mm and $d = 1.5$ mm. **b** The bottom view of the coding element. **c** The transmission schematic for the presented metasurface structure, including four switching schemes with two diodes. **d, e** The reflected magnitude and phase responses of the coding element when the PIN diodes are switched on and off.

bias. The parameters of the sector are $r = 5$ mm and $\beta = 120°$, as shown in Fig. 2b.

The element simulations are performed using the commercial software of CST Microwave Studio. We list the schematic diagram of four working states in Fig. 2c, and the simulated magnitude and phase responses of the reflected waves for different states in Fig. 2d and e, respectively. When the diode along the $x$-axis is turned ON and the one along the $y$-axis is OFF, it represents state "10", and the other three states by parity of reasoning. Given that the structure of the presented element is symmetrical along the $x$- and $y$-axes, the phase modulations for the $x$- and $y$-polarized incidences are the same. Consequently, we give the results only under the $x$-polarization. From Fig. 2d, we observe that the normalized reflected amplitude is almost over 0.8, which guarantees good reflection efficiency at the central frequency of 5.8 GHz (marked in gray). Figure 2e illustrates the phase responses of the element. We remark that the results are for the $x$-polarized incidence, in which the diode along the $y$-axis is kept invariant. From Fig. 2e, it is observed that the OFF−ON states of the $x$-polarization, e.g., the states of "01" and "11" in the top figure and the states of "00" and "10" in the bottom one, have a phase difference of 180˚ around 5.8 GHz.

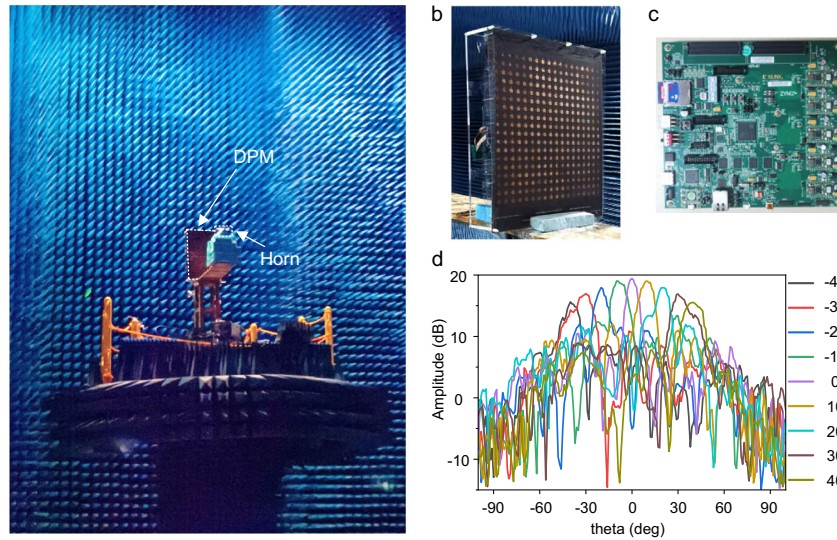

**Fig. 3 | Measurements of the manufactured DPM. a** The far-field experimental setup in an anechoic chamber. **b** Photographs of the fabricated prototype. **c** Zynq-7000 SoC series FPGA for voltage control. **d** The measured far-field patterns when beams on the E-plane vary from −40° to 40° at 5.8 GHz. These experiments verify that the DPM can shape the far-field patterns in the spatial domain by configuring the digital codes.

A total of 324 (18 × 18) elements constitute the aperture of DPM, as shown in Fig. 3. A dual-polarized rectangular horn antenna is used to illuminate the DPM. Based on the superposition principle, the reflected waves of DPM are the superposition of the reflected waves of all elements. By adequately configuring the digital coding scheme, DPM can steer a single beam or two beams dynamically.

## Beam steering by DPM

We consider the specific situation of target recognition and tracking using DPM, and exam the working mechanism of beam steering (see Supplementary Note 2 for details). The experiment for measuring the far-field patterns of DPM is established in a microwave anechoic chamber, as illustrated in Fig. 3a. DPM and the feeding horn are both mounted on a turntable. The feeding horn antenna is used to emit the EM wave with a frequency of 5.8 GHz via a signal generator (Keysight E8267D), and a receiving antenna is used to record the scattered powers via a spectrum analyzer (Keysight E4447A). The feeding horn antenna, with a relatively flat wavefront, is placed 0.5 m away from the metasurface. The photographs of the fabricated sample and details of the biasing line are shown in Fig. 3b and c. Considering the cost and complexity of fabrication, the biasing lines of different polarizations on the bottom layer are printed on two sides of the sample, thereby leading to independent controls of bias voltages. With the aid of the above theoretical analysis, we can achieve precise controls of the scattering patterns at the front space of the metasurface.

As the basis to track moving targets with beams, we discuss the design of the coding scheme for beam steering. Fig 3d plots the measured beams on the E-plane from −40° to 40° with an increment of 10°. The fabricated DPM presents the great performance of dynamic beam scanning controlled by the FPGA shown in Fig. 3c. With the increment of the scanning angle, the gain decreases from 19.43 to 15.54 dB and the beam width becomes wider due to the fact that the effective aperture of DPM becomes smaller when the scanning angle increases. The digital coding schemes and simulation results are presented in Supplementary Note 3. We note that the element itself is symmetrical and the performance of beam steering under the *x*-polarization is good (see Supplementary Note 4 for details). The period of the element is relatively large at 5.8 GHz, and the manufacturing technique has some limitations on the overall size of the metasurface. Hence the metasurface has a relatively small number of 324 elements, leading to the existence of sidelobes in the reflected beam. The

measured results of the dual-polarizations demonstrate that sufficient sidelobe suppression is still held in the measurement, which is sufficient for generating the directive beams. The good performance of designable radiation patterns and spectral power distributions guarantees the feasibility of the proposed intelligent tracking system.

## Platforms for target detection

The RS-Camera is used to collect the target images in real-time at a rate of 40 FPS. The sampled images are then processed by the YOLOv4-tiny network to obtain the real-time positions and poses (i.e., the elevation and azimuth angles relative to the sampling position) of the target. High-precision detection at a certain detection speed is realized by introducing the Mosaic data enhancement, spatial pyramid pool structure (SPPNet), and CSPDarknet53-tiny network with stronger feature extraction ability to the YOLOv4-tiny network, as shown in Fig. 4.

During the inference of the YOLOv4-tiny network, the original image size is firstly scaled to [608, 608, 3] and put into the CSPDarknet53-tiny network for feature extraction. Then the features are fed into two different CNN modules to obtain feature maps with different scales. The feature map with dimensions [38, 38, 256] is obtained after 10 layers of convolution and pooling, and the feature map of [19, 19, 512] is obtained through 4 layers of convolution and pooling. The detection results are obtained by performing 4-layer convolution on the feature maps of [19, 19, 512] to make the number of channels num-anchor×(5 + num-class). On the other hand, the 19 × 19 feature map is doubly up-sampled to a size of 38 × 38. The feature maps of the same size are superimposed in the backbone network to form a new feature map, which integrates the information of the middle and deep layers for better representation ability and then performs 2-layer convolution to change the number of channels. After that, the detection results are obtained in a dimension of num-anchor×(5 + num_class), where num-class represents the number of categories that can be detected, num_anchor represents the number of anchor boxes, and the five parameters represent the center coordinates, width, height, and confidence of the detection box, respectively (see Supplementary Note 5 for details). To be noted, the RS-Camera is not in the center of the aperture of DPM, and therefore the position information obtained by the RS-Camera has a small deviation in the coordinate system of the DPM. We unified the two coordinate systems in the control system so as to make the position information more

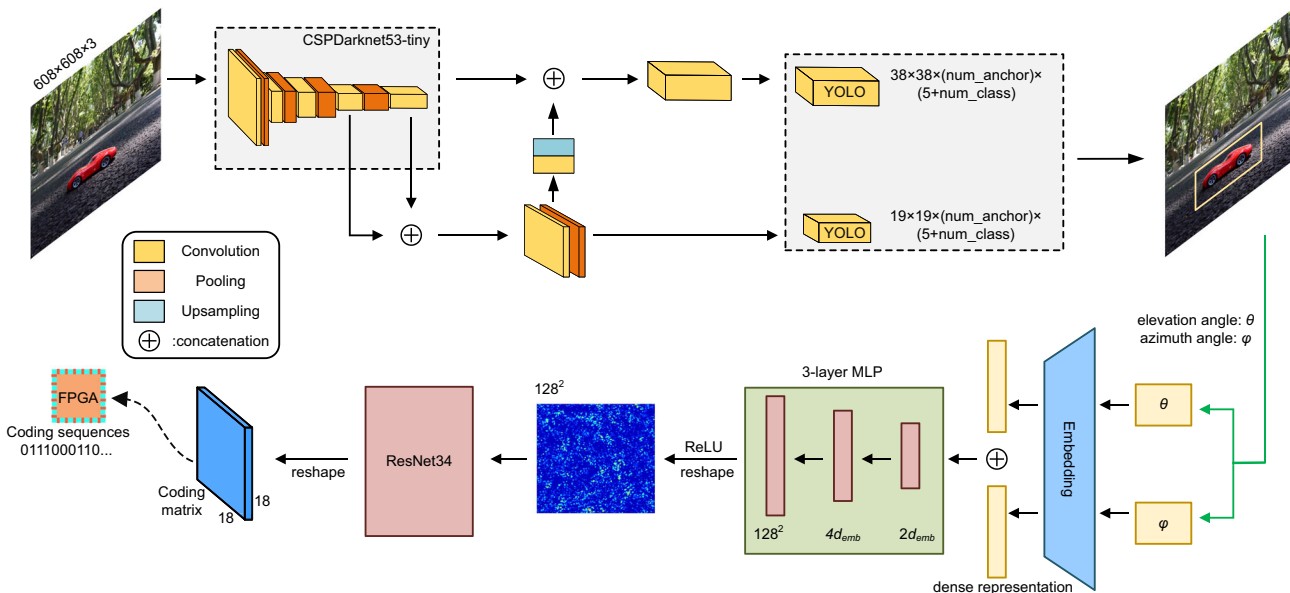

**Fig. 4 | Structure diagrams of the YOLOv4-tiny network and pre-trained ANN.** In the application scenarios of this article, the location of the target needs to be obtained by the RS-Camera, and the pixel coordinates and the label of the target in the image need to be determined first. The input image is fed to a network based on YOLOv4-tiny for feature extraction and target tracking. The detected target information is sent to the pre-trained neural network based on ResNet34 as input. The coding matrix is computed and converted into a coding sequence, and then sent to FPGA for controlling the scattering patterns with complex requirements. More details are provided in Supplementary Notes 6, 7, and 8.

accurate. Through debugging, the position information detected by the RS-Camera and the beam direction of the DPM is precisely consistent. To characterize the working performance of the networks, the detailed structures of the YOLOv4-tiny network and pre-trained ANN are presented in Supplementary Notes 6 and 7, respectively.

To use the pre-trained ANN to decide the corresponding coding sequence, we consider the one-dimensional feature extraction form of the network as a fully connected neural network, and the residual network resnet34 extracts two-dimensional features. The input is the 2D vector obtained earlier, which is composed of the angles of theta and phi, and the output is an $N$-dimensional signal sequence composed of "0" and "1", which is used to control the feeding of metasurface elements. We consider fitting the $N$-dimensional sequence here as a multi-label and multi-classification problem. There are a total of $N$ tags corresponding to $N$ feeds, and each tag has two states corresponding to "0" and "1", respectively. At this time, we take binary cross entropy (BCE)-loss as the loss function. Because the final output has two discrete values of "0" and "1", we use the sigmoid function as the activation function of the last layer of the network. Its value ranges from "0" to "1", and hence can normalize the values calculated by the previous network. After the network training is completed, we set 0.5 as the threshold for judging whether the output is "0" or "1". We denote that the pre-trained ANN can quickly obtain more datasets to cope with the recognition with smaller resolutions. In this study, ANN is designed to learn the coding matrix with low sidelobe characteristics obtained from the particle swarm optimization (PSO) methods, and the output has better beam accuracy with a much faster speed. Therefore, the proposed ANN has stronger abilities than the back-propagation method to solve complex scattering problems and faster speeds than the nonlinear optimization method. For a realistic environment, the presented ANN has the advantages of lightweight, easy deployment, and anti-interference (see Supplementary Note 8 for details).

### Experimental setup and environment

For experimental verification, we fabricated a DPM sample with the size of $470 \times 500 \times 4 \ mm^3$ ($18 \times 18$ elements). The computer vision detection based on YOLOv4-tiny and the pre-trained beam steering

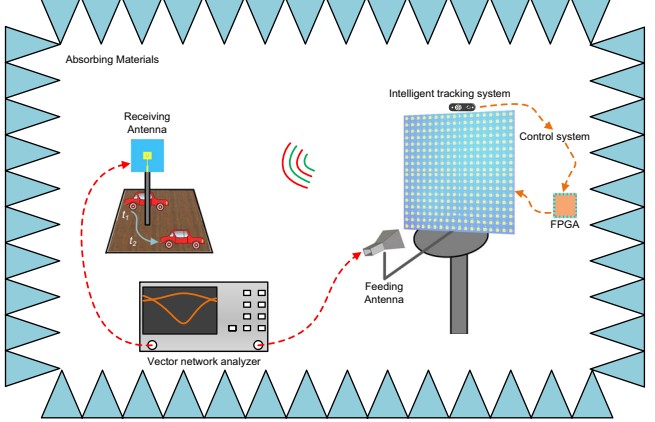

**Fig. 5 | Experimental setup and environment.** A TE-polarized beam from a feeding horn antenna is incident on the metasurface. The position of the moving car is processed in the control system, and all bias voltages are instantly calculated and supplied to the metasurface. The reflected waves of the metasurface are detected in real-time by the receiving antenna.

ANN facilitate the connection between different parts of the system, so as to complete the intelligent tracking. To validate the above concept and method, a target recognition and tracking system is built for experimental demonstration in an indoor scenario.

As shown in Fig. 5, the experiment system consists of the transmitting part, the receiving part, and the moving target. During the experimental test, the EM wave centered at 5.8 GHz is emitted from a feeding horn antenna and reflected by the DPM. The thereby generated radiating signal is then received by a patch antenna attached to the moving target (an electronic model car). The receiving antenna (see Supplementary Note 9 for details) is used as the representation of the moving target. The RS-Camera is used to detect the implementation scene, so as to obtain the car's location in terms of the elevation and azimuth angles. The two angles are then used as the input of the pre-trained ANN, and the coding sequence of DPM is achieved at the

output. The coding sequence is powered to the metasurface by FPGA in the form of voltage. A directive beam towards the moving target is generated by the metasurface under the control of the real-time varying coding sequence to keep tracking the target at a sampling speed of 0.2 s (see Supplementary Note 10 for details). The speed limit lies in the sampling speed of the RS-Camera. We set the speed of 35 frame rates per second to collect the data, but the RS-camera samples the targets every three frames and returns the sampled data only once to the designed ANN. Then the position information and the output voltage sequence are updated and sent to FPGA. In our designed network, the precisions of both elevation and azimuth angles can be as small as 1°. In the practical scenario of small-scale communications with a few users, the increment of $\theta$ and $\varphi$ is set to 3° for demonstration, and the detected angle was rounded to the nearest integer value. In this way, one is able to realize the closed-loop operation of the tracking system and bridge the gap between visual detection and microwave communications.

In order to prove that the system is an adaptive working scene without human intervention, several scenarios are designed in three groups of testing experiments to verify the efficiency and feasibility of the intelligent tracking scheme.

### Moving target detection and identification

We rely on the prototype of a 1-bit dual-polarized DPM to carry out the first group of experiment tests. We first demonstrate the capability of the system to detect the moving model car and track it with directive beams. Fig 5 shows the constituent part of the metasurface-based transmitter that is mainly composed of a vector network analyzer (VNA) and the 1-bit dual-polarized DPM fed by a linearly polarized horn antenna connected to VNA. A patch antenna is carefully designed at 5.77 GHz to serve as the receiving terminal, which is located at the position of the moving target or fixed somewhere in the moving path. We design two experiments to verify the tracking scheme through the two-port VNA, whose input and output are connected to the receiving antenna and feeding horn, respectively. In the first experiment, the receiving antenna is fixed somewhere in the moving path of the car. The reflected beam from DPM is always manipulated towards the car. When the car starts to move, it is far away from the receiving antenna, and the energy received by the antenna (in terms of S21 read in VNA) is very low. As the car moves closer to the receiving antenna, the received energy becomes higher. When the car is the closest to the receiving antenna, the received energy is the highest. After that point, the received energy gradually decreases as the car moves away from the receiving antenna. Please refer to Supplementary Movie 1 for details.

In realistic environments, some complicated scenarios may happen, such as multi-object tracking (MOT), temporarily blocked target tracking, and in limited ambient light. Below we will present the solution of the tracking algorithm in these scenarios. In fact, multiple similar targets and target occlusion are two key problems in the field of target tracking. To solve the problem of similar target interference in the visual field, multiple targets are firstly numbered and the corresponding number of them in the video stream is guaranteed to remain unchanged. And the deep SORT algorithm introduces the Kalman filter to solve the problem of transient target occlusion and the problem of missing individual frame detection. The performance of the object detection algorithm based on the RS-Camera in the above scenarios is presented in Supplementary Note 11 and Supplementary Movies 2 and 3. When multiple targets with different characteristics exist, the YOLOv4-tiny target detection algorithm can classify the targets in the field of vision at the same time, and decide the categories that the targets belong to. By judging the category, the position information of specified target is extracted, and the beam is controlled to point to the specifically tracked target (see Supplementary Note 12 and Supplementary Movie 4 for details). The appropriate upgrades to the hardware in the system are good for more complex scenarios, such as the infrared thermal imagers which also have important applications in industry and temperature monitoring. Therefore, a night version infrared-cut camera (NV-Camera) serves as an aid to solve the detection task under the condition of limited ambient light or completely dark. Experimental results under different light intensities demonstrate that when the light intensity is low, the system can switch from the RS-Camera to NV-Camera to complete the target detection task (see Supplementary Note 13 and Supplementary Movie 5 for details). Based on these experiments, we conclude that the proposed system has the ability to track moving targets, and the target detection algorithm can adapt to realistic environments to complete the detection tasks.

### RF signal detection

Next, we build an RF signal detecting system in the experimental scenario to conduct the real-time tracking scheme for obtaining more intuitive detection, as illustrated in Fig. 6. The transmitter primarily consists of a microwave signal generator (Keysight E8267D), and the DPM fed with the linearly polarized horn antenna. Again, the RS-Camera is placed on the top of DPM. A portable RF signal detector, which consists of a receiving patch antenna, a battery, a detector AD8317, and a microcontroller unit (MCU), is attached to the moving car. Detector AD8317 is adopted to accurately measure the RF signal power in 1 MHz–10 GHz, as illustrated in Supplementary Note 14. It is supplied with 3 V voltage and used to convert the RF input signal to the corresponding dB scale with accurate logarithmic consistency. The battery supplies powers to MCU (Arduino), and the DC port on the MCU supplies power to the detector AD8317. The input of the detector AD8317 is connected to the receiving antenna, and the output is connected to MCU for monitoring and processing in real-time. In this way, the portable detector without an additional voltage source is realized.

To demonstrate the validity of the intelligent metasurface system, we have designed two experimental demonstrations in an anechoic chamber and outdoor scenarios. Firstly, the detector is placed in the middle of the moving path of the car inside an anechoic chamber, together with the receiving antenna. We observe that the received signal increases as the car comes closer and then decreases when the car goes away, as shown in Fig. 6b. Secondly, the portable RF signal detector is attached to the car to observe the change of RF signals during the movement. We have collected the data sets for the detector-loaded car so that the RS-camera can correctly capture the moving target in the identification process. We monitor the recognition in this experiment, as seen in Supplementary Movie 6. Four curves in Fig. 6b, respectively, plot the voltage values obtained by the detector and the corresponding dB calibration values. We observe that when the detector is fixed in the middle of the moving path, the received energy has the highest value when the car is closest to the detector. In contrast, when the detector and the car move together, the received energy is relatively stable with a high value.

Then, we carry out experiments in an outdoor environment to conduct the real-time tracking scheme in dual polarizations. The testing sites are chosen at the campus of Southeast University. Figure 6c shows the outdoor environment and experimental setup, including the transmitting horn, RF signal detector, DPM locations, signal generator, FPGA module, and control system (see Supplementary Note 15 for details). This field trial is conducted outside the side entrance of our laboratory. The portable RF signal detector is attached to the car to observe the change of RF signals during the movement. Four curves in Fig. 6d plot the voltage values obtained by the detector and the corresponding dB calibration values under dual polarizations, respectively. When the detector and the car move together, the received energy is relatively stable with a high value. This number is well aligned with the power gain observed in the indoor test.

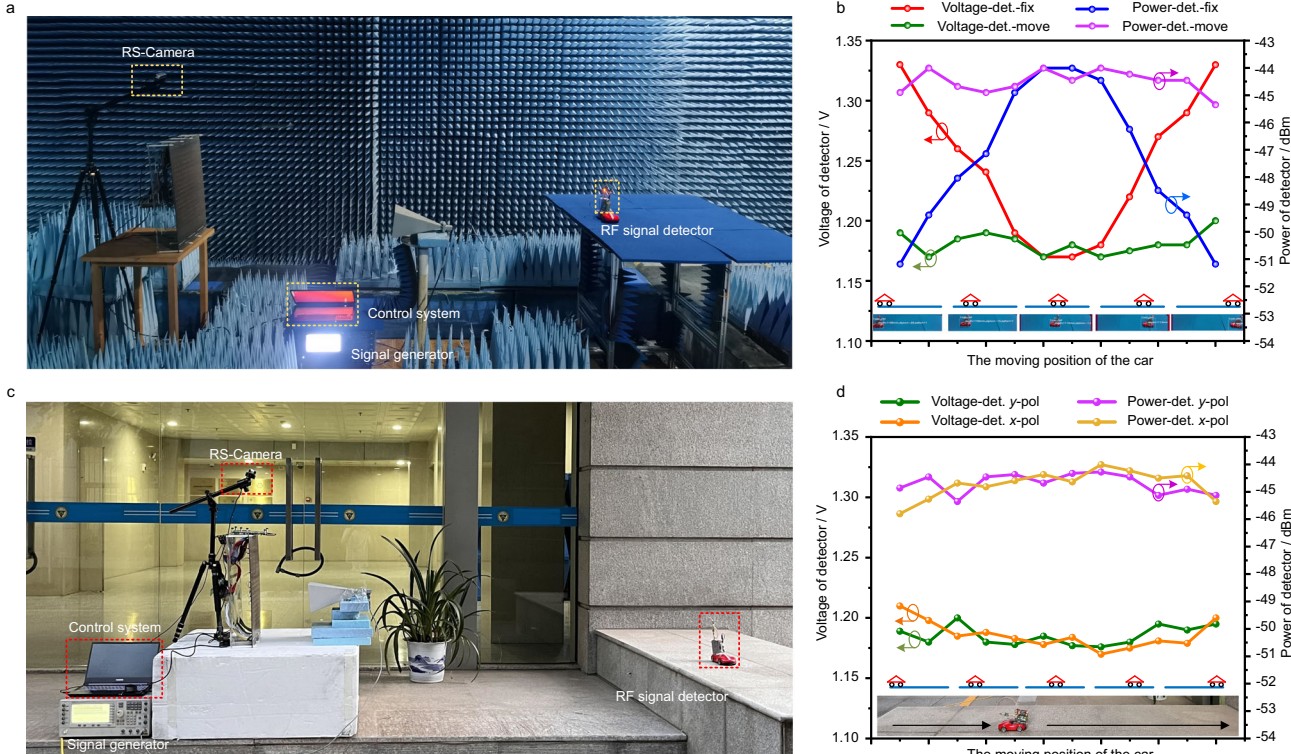

**Fig. 6 | Experiments of radio-frequency (RF) signal detections. a** A TE-polarized beam from a feeding antenna is incident on the DPM. The position of the model car is fed into the control system, and all bias voltages are instantly calculated and supplied to DPM. The reflected waves are detected in real-time by the RF signal detector. **b** RF signal changes when the detector is fixed (Voltage-det. -fix) in the middle of the path or moves (Voltage-det. -move) with the car. **c** Outdoor experimental setup of the intelligent metasurface system. **d** The received RF signals under *y*-polarization (*y*-pol) and *x*-polarization (*x*-pol) when the detector moves with the car. The horizontal ordinate is the moving path of the car from the beginning to the end.

With respect to communications in a more practical environment, the interference at 5.8 GHz in realistic environments could be high, and the designed DPM can solve the potential interference by its own characteristics and attached sensing devices. More discussions are illustrated in Supplementary Note 17. Through these experiments, we prove that the scheme is real-time and can be quantitatively verified by the RF signals.

### Real-time wireless transmissions

Aside from intelligent tracking, we further demonstrate that the scheme has the power to realize high-speed transmissions of information with the moving target. Here, we present two experiments of real-time video transmissions for instance, as demonstrated in Fig. 7a and b. The video is taken by the camera to capture the change in screen and is sent to the video module which serves as a wireless image transmission module in the frequency range from 5.65 to 5.95 GHz. We remark that the working frequency bands of the video module, DPM, and the 5GHz-Wi-Fi all include 5.8 GHz. In the first experiment, the receiver is still placed in the middle of the path, and the video is transmitted only when the car moves close to the receiver (see Supplementary Movie 7 for details).

As shown in Fig. 7c, during the experiment, we select five different positions of the car on the moving path for demonstration. The bit error rate of video information transmission can be seen intuitively. When the car is far away from the receiver, the error rate is high and the receiver cannot receive the video information. When the car moves near the receiver, the video can be transmitted clearly. In the second experiment, the receiver is attached to the car, and the video is always transmitted smoothly while the car is moving (see Supplementary Movie 8 for details). As shown in Fig. 7d, we intercepted five different

positions in the experimental process when the receiver and the moving car are bound together, and the captured video kept transmitting at a low bit error rate. To sum up, in these two experiments, the effect of wireless transmission is demonstrated by intercepting five states of the transmitted images, as shown in Fig. 7c and d. When the receiver is fixed in the middle of the moving path, effective wireless transmissions can be realized only when the car is close to the receiver and the beam is manipulated towards them. In contrast, when the receiver is attached to the car, the movement of the car does not affect the transmission of video because the beam is dynamically tracking them. Here, the vector signal transceiver (VST, PXIe-5841, National Instruments Corp.) is used for the bit error rate (BER) measurement, in which we set the modulation mode as QPSK and the transmission rate as 170 Mbps. When the channel is in the acceptance region, the BER is stable at $10^{-5}$ (see Supplementary Note 16). Moreover, in a more practical environment, there will be interference problems such as other communication devices in adjacent frequency bands. The programmable ability and dual-polarization performance of DPM itself, help to eliminate the interference together with wireless sensing and other potential ways (see Supplementary Note 17 for details). We estimate the energy consumption of the proposed design in Supplementary Note 18, including object detection, communication, and power supplies. The performance of the design under different input powers is also discussed and tested, as presented in Supplementary Note 18.

### Discussion

For the intelligent system and wireless communications, we proposed a scheme of target recognition and tracking systems based on DPM and computer vision. In the intelligent system, the RS-Camera

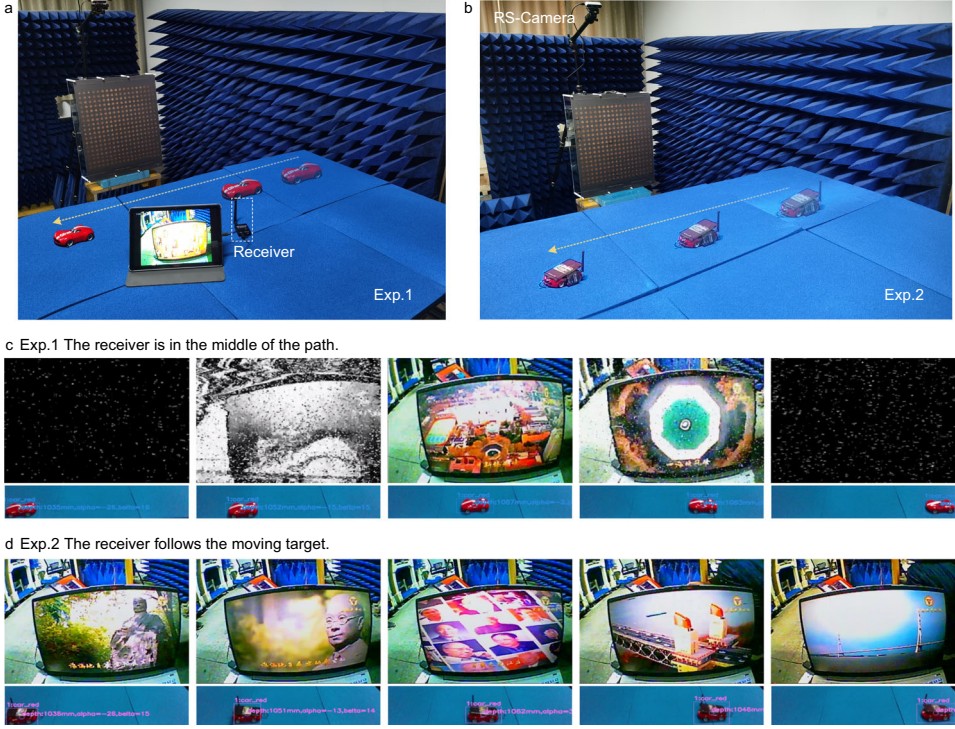

**Fig. 7 | Experiments of wireless communications. a, b** Experiment scenarios when the receiver is in the middle of the path (**a**) and when the receiver follows the moving target (**b**). **c, d** Experimental results of received video frames at five different positions of the car on the moving path in (**c**) Experiment Scenario 1 and (**d**) Experiment Scenario 2.

combined with YOLOv4-tiny is used to detect the position information of the moving target, and process the detected information using a pre-trained ANN to obtain the required coding sequence for the voltage control of DPM. Then intelligently adaptive beams are generated by DPM to track the moving target in real-time. This system runs effectively in a closed loop without human intervention. Experimental verifications have been carried out in different scenarios, proving that the proposed scheme has the capabilities of intelligent tracking and information transmissions. This may find promising applications in other intelligent and self-adaptive systems, including intelligent and multi-physical sensing, and the internet of things (IoT) technologies. The proposed concept of intelligent tracking metasurface will also open up an avenue for challenges in the 5G and 6G wireless communications, enriching the functions of metasystems.

## Methods
### Details on the digital programmable metasurface
The 1-bit dual-polarized DPM used in this work operates around the central frequency of 5.8 GHz and contains $18 \times 18$ programmable elements. It is designed in the commercial software CST Microwave Studio and fabricated with printed circuit board technology. Metallic structures on the top and feeding circuits on the bottom are printed on the commercial dielectric substrate F4B with dielectric constant $\varepsilon_r = 3$ and tangent loss $\tan \delta = 0.003$. The photographs of fabricated prototypes are shown in Fig. 3. Two PIN diodes (SMP1320 from SKYWORKS) are embedded into each element to control the reflection phase.

### Measurement setups
The experimental setup for measuring the far-field patterns were established in a microwave anechoic chamber, as illustrated in Fig. 3. The DPM and the feeding antenna were both mounted on a turntable. A feeding horn antenna was used to emit the monochromatic carrier wave with frequency $f = 5.8$ GHz via a signal generator (Keysight

E8267D), and a receiving antenna was used to record the scattered powers via a spectrum analyzer (Keysight E4447A).

In the experimental process of moving target detection and identification, a transmitting horn antenna, a receiving patch antenna, a DPM, and a vector network analyzer (VNA, Agilent N5230C) are set up in the anechoic chamber, as shown in Fig. 5. The moving target is represented by one or two model cars, which are captured by the RS-Camera. VNA is used to acquire the response data by measuring the transmission coefficients ($S_{21}$). To suppress the noise level in measurement, the intermediate bandwidth in VNA is set to 40 MHz.

During the experiment of RF signal detection, the detector based on AD8317 and MCU is used to measure the receiving level. The transmitter consists of a microwave signal generator (Keysight E8267D) and the DPM is fed with a horn antenna. During the test, voltage values are obtained by the detector, and the output voltage value is connected to the MCU to obtain the corresponding dB scale for real-time monitoring and processing.

In the experimental process of real-time wireless transmission, the image transmission module collected a picture of a video played through the notebook and sent it to the horn antenna after modulation. The video information is transmitted to the DPM through the horn antenna, and then to the receiving module. The receiving module is a receiving antenna, a decoder, and a screen connected to the decoder. When the information can be accurately transmitted, the screen will restore the image collected by the image transmission module. The experiment scenarios are shown in Fig. 7, in which the video transmission module and the feeding horn are located under the supporting platform and the horn is about 0.8 m away from the DPM.

## Data availability
The authors declare that all relevant data are available in the paper and its Supplementary Information Files, or from the corresponding author on request.

## Code availability

The custom computer codes utilized during the current study are available from the corresponding authors on request.

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

## Acknowledgements
This work was supported by the Basic Scientific Center of Information Metamaterials of the National Natural Science Foundation of China (62288101, T.J.C.), the National Key Research & Development Program of China (2017YFA0700201, 2017YFA0700202, 2017YFA0700203, T.J.C., 2021YFB3200502, W.T.), the National Natural Science Foundation of China (61971134, W.T., 61631007, T.J.C., 62101589, T.Q.), the Major Project of the Natural Science Foundation of Jiangsu Province (BK20212002, T.J.C.), the Fundamental Research Funds for Central Universities (2242021R41078, W.T.), and the 111 Project (111-2-05, T.J.C.).

## Author contributions
W.L., W.T., and T.J.C. conceived the idea, conducted the theoretical analysis, and wrote the paper. W.L., Q.M., and C.L. proposed the concept of digital programmable metasurfaces and pre-trained ANN and built the proof-of-principle prototype system. W.L., C.L., Y.Z., X.W., Jiawei W., S.G., T.Q., T.L., Q.X., T.T.G, Z.Z., F.L., and Jiaxuan W. conducted experiments and data processing. T.J.C., Q.C., and L.L. provided suggestions and comments and helped to organize and revise the draft. All authors discussed the results and contributed to the manuscript.

## Competing interests
The authors declare no competing interests.
