## [Peer Review File · Nature Communications]

Intelligent metasurface system for automatic tracking of moving targets and wireless communications based on computer visionReviewers' Comments:

Reviewer #1:

Remarks to the Author:

This manuscript proposes a design of digital programming metasurfaces to track moving targets and perform wireless communications at 5.8 GHz using computer vision and machine learning algorithms. The authors first present several existing approaches related to target tracking, such as radars, and point out their deficiencies when dealing with mobile users. The authors also discuss the limitation of recent metasurfaces that lack self-adaptiveness and need external control or human intervention to achieve certain functionalities. The authors then discuss their proposed design with experimental results conducted in an anechoic chamber under different scenarios.

To the best knowledge of this reviewer, this work is the first one combining computer vision, machine learning algorithms, and digital coding metasurfaces to perform real-time data transmission and target tracking, so it demonstrates novel results. The design of DPM, field pattern simulation, experimental configurations as well as relevant results are demonstrated and explained in a well-organized manner. This manuscript has a good flow and a clear structure.

This reviewer has the following comments for the authors to further revise their work:

1. The authors provide a comprehensive discussion on target detection using the Yolov4-tiny network based on sample images collected from the RS camera. The experiments show at most two model cars being used for the model validation. It would be better if the authors could also briefly discuss the complexity of the proposed target detection model from a mathematical perspective. For example, the reasons for choosing the parameters used in their detection algorithm. In the current supplementary notes S1 and S4, it is not straightforward to locate such information.
2. It seems that an underlying assumption for target tracking using computer vision algorithms presented in this work is that the default setting always has good ambient light. However, in some realistic scenarios, the environment could be dark. How does the RS-camera-based target detection algorithm perform when there is limited ambient light, or the environment is completely dark?
3. The experiments were conducted in an anechoic chamber, which is considered an ideal lab environment. However, in realistic environments, including indoor and dense urban outdoor scenarios, both the target tracking as well as communication might not be as simple as shown in the results. For example, there might be multiple similar objects present with the target, or the target might be temporarily blocked by other objects. In this case, how would the tracking algorithm perform? It would be better if the authors could discuss this issue.
4. With respect to communications in a more practical environment, the interference at 5.8 GHz in realistic environments could be high, since this is an ISM band. How does the designed DPM solve the potential issue of interference from other actively communicating devices operating at a similar frequency?
5. In the DPM design, what is the role of dual polarization in experiments? The experiments seem not to take advantage of the polarization diversity. It would be better if the authors could also discuss this matter.
6. In the 2D vector that serves as the input to the ANN, what is the precision of the angles? Are they rounded to the nearest integer values to follow the 10-degree increment of the beams from the DPM?
7. In Line 272 under subsection "Experimental setup and environment", it is not clear what the term "every three times" means, if it is three seconds or three time samples. It would be better if the authors could further clarify this.

8. In the subsection of "RF signal detection", how are the voltage values measured by the detector converted to the power values? It would be better for the authors to present more details.

9. In the subsection of "Real-time wireless transmissions", the bit error rate performance needs quantitative analysis, in addition to the photos shown in Fig. 7, to help readers with a better understanding.

10. The energy consumption of the proposed design, including target detection, tracking, and communication, is another aspect that might affect the system performance. Some numerical results would be better to understand this matter.

Reviewer #2:

Remarks to the Author:

Please find my comments in the attached pdf file.

Review of Intelligent metasurface system for automatic tracking of moving targets and wireless communications based on computer vision

This paper presents the development of a system allowing an active reconfiguration of a metasurface whose radiation is controlled by a convolutional neural network. This work is conducted in an incremental framework based on physical solutions introduced in previous publications. This system is fed by optical data and facilitates beamforming towards a target in an ideal electromagnetic environment (anechoic chamber). The aimed applications are oriented towards target tracking and wireless communications, in a context motivated by the increasing development of reconfigurable means and allowing the possible increase of the transferred data rates.

Two points seem to me currently blocking to recommend the publication of this paper in Nature Communications.

1. The bibliography is almost entirely oriented towards the work of the authors or their direct collaborators. The presentation of the numerous applications of these new technologies in the introductory part thus eludes in part or completely the contributions of the many research groups which however contributed to found some of the cited applications.

Following an extraction of all the papers cited, the figure inserted in the following page presents a graph of interaction between authors. The latter allows to highlight the very high rate of self-citation of some of the contributors of the paper I have been charged to evaluate.

Insofar as the positioning of this work appears almost absent in comparison with numerous pioneering contributions proposed by other research groups, the authors' work does not allow the reader who is not familiar with this field to form an objective opinion of the real contribution of this paper to the literature.

2. The proposed demonstration is convincing but the complexity of the proposed approach in such a controlled environment does not seem justified.

To summarize the proposed approach, the principle relies first on the optical detection of the angle of a target with respect to the electromagnetic surface reference frame. Beamforming is then performed from the latter using phase modulations coded on 1bit according to patterns that can be readily justified analytically.

What is the added value of a technique based on the exploitation of artificial neural networks compared to solutions that are simpler to implement, potentially faster and where physical hindsight is not abandoned in favor of machine learning solutions? It is necessary that the authors can justify the usefulness of the proposed approach, especially in experimental conditions as controlled as in an anechoic chamber.

Figure 1 - Interaction graph of authors extracted from the bibliography of the evaluated paper, highlighting a very high rate of self-citation and the absence of contributions from active/pioneer groups among the applications studied in the introduction of this paper.

Reviewer #3:

Remarks to the Author:

This manuscript presents an intelligent tracking system based on the digital programmable metasurface and computer vision. The authors use computer vision to locate the target to be tracked and send the position information to the intelligent metasurface system so as to realize self-adaptive beam tracking of the moving target. Experiments of radio frequency signal detection and wireless information transmission demonstrated the function of the proposed system and showed good stability. In my opinion, this work is interesting and the proposed intelligent tracking method can provide important and practical help for self-adaptive and smart communication systems. I can recommend this paper for publication after minor revision. Below are specific comments to be addressed.

1. The authors put forward a 1-bit dual-polarized digital programmable metasurface, which seems to be controlled independently by each unit. However, the reconfigurable scheme of units and configuration of feeding network are not clearly explained. Please give information on these issues, and also clarify the bandwidth of the reconfigurable metasurface.
2. In the intelligent tracking system, the RS-camera serves as the auxiliary to complete the task of moving target identification and tracking. What is the volume (width×depth×height) of the real scenario captured by the RS camera, and how does the system discriminate and switch between multiple targets?
3. On page 14, line 339, it is stated that "We collect the data sets for the detector-loaded car, so that the RS-camera can correctly capture the moving target in the identification process." The authors need to show the collection and processing of data sets.
4. I noticed that the SI provides the experiment on the response speed of the FPGA and the intelligent tracking system by logic analyzer. Is it possible to achieve higher frame rates using the methods presented in this article? Please clarify.
5. In the sub-sections of "Moving target detection and identification" and "RF signal detection" there exist some similarities in the description of experimental results. Please increase the readability and conciseness of the article.

Response Letter to Reviewers

We are grateful for the constructive comments on this manuscript (NCOMMS-22-20669) from all the referees. In the text below, each comment is quoted in italics and is followed by the corresponding detailed response. We have also revised the manuscript and the supplementary material accordingly, and highlighted all changes in the revised documents.

General comments from Referee #1:

This manuscript proposes a design of digital programming metasurfaces to track moving targets and perform wireless communications at 5.8 GHz using computer vision and machine learning algorithms. The authors first present several existing approaches related to target tracking, such as radars, and point out their deficiencies when dealing with mobile users. The authors also discuss the limitation of recent metasurfaces that lack self-adaptiveness and need external control or human intervention to achieve certain functionalities. The authors then discuss their proposed design with experimental results conducted in an anechoic chamber under different scenarios.

To the best knowledge of this reviewer, this work is the first one combining computer vision, machine learning algorithms, and digital coding metasurfaces to perform real-time data transmission and target tracking, so it demonstrates novel results. The design of DPM, field pattern simulation, experimental configurations as well as relevant results are demonstrated and explained in a well-organized manner. This manuscript has a good flow and a clear structure.

Authors Response:

We thank the referee for the positive comments. The insightful comments are very constructive for further improvement of this work. In the following, we address the specific comments point-by-point whilst revising our manuscript.

Specific comments from Referee #1:

Referee #1 -- Comment 1:

1. The authors provide a comprehensive discussion on object detection using the YOLOv4-tiny network based on sample images collected from the RS camera. The experiments show at most two model cars being used for the model validation. It would be better if the authors could also briefly discuss the complexity of the proposed object detection model from a mathematical perspective. For example, the reasons for choosing the parameters used in their detection algorithm. In the current supplementary notes S1 and S4, it is not straightforward to locate such information.

Authors Response:

We thank the referee for this valuable comment. In the object detection algorithm, there are a series of index parameters, which are very important for understanding and evaluating the performance of the algorithm, and we agree that it is necessary to describe the object detection algorithm in more details to enrich the manuscript.

To your question, we mainly made the following four parts as supplementary explanations:

- (1) The reason for selecting the object detection, and the YOLO series in the detection algorithm.**
- (2) Design of loss function in YOLOv4-tiny.**
- (3) Structure of YOLOv4-tiny (presented as a Supplementary file).**
- (4) Meaning and values of some important hyper-parameters in YOLOv4-tiny.**

In computer vision, there are four main types of tasks regarding image perception: **classification, localization, detection and segmentation**. Classification is responsible for determining the class of a target contained within an image, localization is responsible for determining the location of the pixels of the target, detection includes locating all targets in the image and classifying them, and segmentation requires determining the target or scene to which each pixel belongs.

Fig. R1 Schematic representation of four main tasks in computer vision.

Object detection is an overlay of classification and regression. In the application scenarios of this work, the location of the target needs to be obtained by the vision sensor, and the pixel coordinates and the label of the target in the image need to be obtained first. In view of this, object detection in the field of computer vision is well suited in this work.

In the early visual object detection, people used sliding windows of various sizes to slide on the image to select candidate regions. Manually select which features to extracted from the candidate regions, to determine the presence or absence of targets as well as the classes of targets in the candidate boxes, as shown in Fig. R2. However, such detection methods are

time-consuming and the manually designed features are poorly robust, difficult to cope with different scenarios.

In view of this, we adopted the YOLO (You Only Look Once) series of algorithms which is a deep learning based on object detection algorithm. There are two main metrics for evaluating the complexity of object detection algorithms, FLOPs (Floating Point Operations) and parameters.

Floating-point operations refer to the number of additions and multiplications performed during the inference of the model, which describes the computational power required by the network model to inference and reflects the performance requirements of the algorithm to the hardware (e.g. GPU). (2) Parameters refers to the number of convolution kernel weights, full connected layer weights and other learned weights in the neural network, which reflects the amount of memory required by the model for inference.

For a convolutional layer, the FLOPs and parameters are calculated as:

$$\text{FLOPs} = HWC_{out}(K^2C_{in} + 1)$$

$$\text{Params} = C_{out}(K^2C_{in} + 1)$$

where 1 denotes the bias operation, H and W are the size of the output feature map, and C_{in} , C_{out} denote the number of input and output channels, and K is the convolution kernel size¹.

The YOLO family of algorithms is currently available in several versions²⁻⁵. Among them, YOLOv4 and derivative algorithms are the most effective. YOLOv4 has 64.4 M parameters and 142.8 GFLOPs, while YOLOv4-tiny has only 6.1 M of parameters, ten times less than YOLOv4, and 6.9 FLOPs (note: 1 GFLOPs = 109 FLOPs). Although the mean Average Precision (mAP) on the COCO dataset was 62.8% for YOLOv4 and 40.2% for YOLOv4-tiny, **the YOLOv4-tiny network with fewer parameters, faster loading and higher speed** was chosen considering our simpler experimental scenario and lower hardware performance⁶. The YOLOv4-tiny algorithm is used in this paper.

Fig. R2 Traditional object detection method

The YOLOv4-tiny algorithm is a supervised learning algorithm that requires images containing targets to be acquired and manually labelled in advance, and used as a dataset to train and validate the model. YOLOv4-tiny takes a colour image consisting of three channels of RGB as input, extracts features through a convolutional neural network, classifies and regresses the image features, and outputs a rectangular bounding box (x, y, w, h) containing the target, the confidence of containing the object in bounding box, and the object class (label1, label2, ..., labelN). In the training mode, all parameters within the convolutional neural network are randomly initialised, and after reasoning on the input image, the output $(x, y, w, h, \text{confidence}, \text{label1}, \text{label2}, \dots, \text{labelN})$ is obtained and substituted into the loss function

with the ground truth to find the loss value and use the gradient back-propagation using the gradient descent method, so that the network parameters converge to the optimal value. The loss function is shown in Equations (1)-(5).

$$L_{total} = L_{xy} + L_{wh} + L_{cof} + L_{cla} \quad (1)$$

$$L_{xy} = \lambda_{coord} \sum_{i=0}^{S^2} \sum_{j=0}^B l_{ij}^{obj} [(x_i - \hat{x}_i)^2 + (y_i - \hat{y}_i)^2] \quad (2)$$

$$L_{wh} = \lambda_{coord} \sum_{i=0}^{S^2} \sum_{j=0}^B l_{ij}^{obj} [(\sqrt{w_i} - \sqrt{\hat{w}_i})^2 + (\sqrt{h_i} - \sqrt{\hat{h}_i})^2] \quad (3)$$

$$L_{cof} = \sum_{i=0}^{S^2} \sum_{j=0}^B l_{ij}^{obj} (c_i - \hat{c}_i)^2 + \lambda_{noobj} \sum_{i=0}^{S^2} \sum_{j=0}^B l_{ij}^{noobj} (c_i - \hat{c}_i)^2 \quad (4)$$

$$L_{cla} = \sum_{i=0}^{S^2} \sum_{c \in \text{classes}} (p_i(c) - \hat{p}_i(c))^2 \quad (5)$$

In equation (1), L_{total} represents total loss, L_{xy} represents loss of centre, L_{wh} represents loss of width and height, L_{cof} represents loss of confidence and L_{cla} represents loss of classification. In equation (2), S^2 indicates the number of anchor boxes, B indicates the number of prediction boxes, $B=3$ in YOLOv4-tiny, l_{ij}^{obj} indicates whether there is a object in the box, x and y indicates the ground truth centrecoordinate, \hat{x} and \hat{y} indicates the prediction box centre coordinate, λ_{coord} indicates the loss of centre coordinate weight ($\lambda_{coord} = 5$). Equation (3) where w denotes the width of ground truth, h denotes the height of ground truth, \hat{w} denotes the width of prediction box and \hat{h} denotes the height of prediction box. In equation (4) λ_{obj} denotes the non-object weight ($\lambda_{obj} = 0.5$), c denotes whether the anchor box contains the object, i.e. $c = 1$ means the ground truth contains the object, $c = 0$ means the opposite, and \hat{c} denotes whether the anchor box contains the predicted value of the target, also means the confidence. In equation (5), $p(c)$ denotes the ground truth of the labelclassification and $\hat{p}(c)$ denotes the prediction for the labelclassification.

In this paper, the YOLOv4-tiny input data is a RGB image of dimension (608,608,3), where (608,608) is the resolution of the image and 3 refers to the colour image consisting of three channels of RGB (red, green, blue). The input data through the convolutional neural network to finally output (19,19,24), (38,38,24), where (19,19), (38,38) is the width and height of the output data, and 24 is obtained from $(5+3) \times 3$, where $\times 3$ means that each point on the output data will correspond to 3 prediction bounding box, and $(5+3)$ represents the prediction box(x, y, w, h) and confidence, predicted label(car_red, car_blue, car_night), if the target in the box is a red car, then the predicted label is (1,0,0) and the blue car is (0,1,0), the predicted label under infrared imaging is (0,0,1).

The structure of the YOLOv4-tiny network used in this paper is shown in supplementary materials. (see Supplementary file YOLOv4-tiny-cfg.svg for details). The YOLOv4-tiny model uses gradient descent to find the minimum of the loss function during

training, which is iterative, meaning that the data needs to be computed several times during training to find the optimal solution. If the training data is too large for all the data to be fed into the computation at once, a small amount of data needs to be put in several times, and the amount of data to be put in each time is the batchsize. The choice of batchsize is crucial, and Fig. R3 shows the training results when the batchsize is of different sizes. In blue, all the data is fed into the training, i.e. the batchsize contains all the training samples. Green is minibatch, i.e. all the data is divided into several batches, each containing a small number of training samples. In red, the training is random, i.e. batchsize=1. As can be seen from the diagram, the best results are obtained by putting the whole data in at once, and this is the best way to train when the amount of data is small and the computer can carry it. If you put in a small number of training samples at a time, there is a slight loss of accuracy, and if you choose a random sample for training, the model will be easily biased by the noise in the dataset, making it difficult to reach convergence. The choice of batchsize therefore also determines the accuracy of the fit of the network model to the training data.

Fig. R3 The training results when the batchsize is of different sizes⁷.

Within a certain range, the larger the batchsize, the more accurate the direction of descent is and the less training oscillation it causes. However, after the batchsize increases to a certain level, the determined descent direction basically does not change anymore, and instead the convergence of parameter is slowed down by the need to process too much data in one iteration. Therefore, the parameter batch=64 in the YOLOv4-tiny paper is chosen in this paper.

For the Nvidia GTX 1650 GPU used in our system, the video memory cannot load 64 images at once, so the subdivisions parameter is used. Let subdivisions=16, which means that the video memory is loaded 4 images at a time, and the results are saved after processing until 64 images have been processed, then gradient descent is performed on their computed loss values. subdivisions ensures that algorithm training can be performed on devices with low computing power.

The max_batches parameter refers to the maximum number of iterations for training, and the algorithm stops when the number of iterations reaches max_batches. max_batches is too small and the model will stop early before it reaches the optimal parameters, while

max_batches is too large and the loss function has already reached its minimum value, and further training will waste equipment resources. The number of images in the training set used in this paper is 162, so max_batches = 9999 can be used to train the model to the optimum.

It is also important to choose the step size of the gradient descent method, as too large a step size can lead to oscillations and difficulty in convergence after iterating around the optimum point, while too small a step size can lead to slow iterations and take longer to converge. Therefore, using a variable step size for parameter update, the training set has 162 data, when iterating 1300 batches, it is equivalent to 500 iterations of all the data, at this time, the update step size can be changed to 1/10 of the original one, so that it converges more finely and it is easier to reach the optimal point. Considering the number of data sets and the training speed of the YOLOv4-tiny model, the step size was reduced to 1/10 of the original size after 1300 and 1800 generations respectively in this paper.

The input size of the YOLOv4-tiny model is fixed and generally chosen as a multiple of 32. In the YOLOv4-tiny paper, two input sizes were used for experimentation; a larger input size would yield more features and higher accuracy, so the input size of 608 was chosen for this paper.

In addition to the above parameters, the hyperparameters that need to be determined for model training are momentum, decay, angle, saturation, exposure, hue, and classes. the meanings and values of the parameters are shown in Table R1.

Table R1 Description and value of some parameters in YOLOv4-tiny

	Implications	Value in this paper
batch	The number of images used for each parameter update during training. When the predicted values are obtained by inference on the batch images, and the loss function and gradient are calculated, then the model parameters are updated using gradient descent.	64
subdivisions	When setting the parameters, the performance of the GPU is taken into account. GPUs with different performance have different video memory sizes, and GPUs with smaller video memory cannot put in batch images for training at the same time, so the batch is divided into subdivisions and put into the video memory for training, but the parameters are still updated once for each batch image.	16
max_batches	Maximum iterations	9999
steps	Learning rate change step	1300, 1800
scales	Learning rate change factor, when the number of iterations reaches steps, the current learning rate is	0.1, 0.1

	multiplied by the scales as the new learning rate.	
Width height	Image size for network input	608, 608
momentum	Momentum parameters in the momentum gradient descent algorithm	0.9
decay	Weight decay canonical coefficients, which are used to prevent over-fittingc	0.0005
Angle		3
Saturation Exposure Hue	Data enhancement parameters applied to the input image during training	1.5 1.5 0.1
classes	Number of target classes to be detected by the network	3

1. Molchanov, P., Tyree, S., Karras, T., Aila, T. & Kautz, J. Pruning Convolutional Neural Networks for Resource Efficient Transfer Learning. *CoRR* **abs/1611.06440**, (2016).
2. Redmon, J., Divvala, S., Girshick, R. & Farhadi, A. You Only Look Once: Unified, Real-Time Object Detection. in *2016 IEEE Conference on Computer Vision and Pattern Recognition (CVPR)* 779–788 (2016).
3. Redmon, J. & Farhadi, A. YOLO9000: Better, Faster, Stronger. in *2017 IEEE Conference on Computer Vision and Pattern Recognition (CVPR)* 6517–6525 (2017).
4. Redmon, J. & Farhadi, A. YOLOv3: An Incremental Improvement. *CoRR* **abs/1804.02767**, (2018).
5. Bochkovskiy, A., Wang, C.-Y. & Liao, H.-Y. M. YOLOv4: Optimal Speed and Accuracy of Object Detection. *CoRR* **abs/2004.10934**, (2020).
6. Wang, C.-Y., Bochkovskiy, A. & Liao, H.-Y. M. Scaled-YOLOv4: Scaling Cross Stage Partial Network. *CoRR* **abs/2011.08036**, (2020).
7. Geron, A. *Hands-on Machine Learning with Scikit-Learn, Keras, and TensorFlow*, 2nd Edition. (2019).

In the revised documents, we added a new Supplementary Note S6 on generalizations of our concepts, which includes the reason for selecting the YOLO series in the detection algorithm, meaning and values of some important hyper-parameters in YOLOv4-tiny.

We also added a new Supplementary file YOLOv4-tiny-cfg.svg to illustrate the structure of the YOLOv4-tiny network used in this paper.

Referee #1 -- Comment 2:

2. It seems that an underlying assumption for target tracking using computer vision algorithms presented in this work is that the default setting always has good ambient light. However, in some realistic scenarios, the environment could be dark. How does the RS camera-based object detection algorithm perform when there is limited ambient light, or the environment is completely dark?

Authors Response:

We thank the referee for pointing out this practical problem on the object detection algorithm performance under limited ambient light. We agree that the issue on light levels should not be ignored.

In the computer vision (CV) tasks based on image vision, different light intensities are often encountered, which affects the contrast of images and consequently the final result of CV tasks. In many realistic scenarios, due to the lack of robustness of computer vision, multi-sensor fusion should be paid more attention in future research and practical applications. Visual scene understanding in complex scenarios is a problem that must be solved through combining computer vision tasks with applications (such as the unmanned driving and robots). Therefore, in complex environment, it is necessary to optimize and upgrade the hardware and software of a variety of new imaging research technologies. For example, the infrared thermal imager, which works in dark environment, not only has a wide application value in the industrial field, but also have an important application in epidemic prevention and public safety. Hardware synchronization and software integration of infrared image and visible image will make it easier to solve the problem of object detection in limited ambient light.

In order to answer this question in a more straightforward way, we use night vision (NV-) infrared-cut camera as an aid to solve the detection task under the condition of insufficient light and darkness. In the experiment, a digital photometer was used to test the light intensity, and the digital photometer was placed in the lower right corner of the test scene. In order to control the reflection of light on the floor of the test platform, please allow us to place absorbing cotton on the platform. First of all, using the existing experimental equipment in the manuscript, we carried out eight experiments to test the object detection algorithm under different light intensities. Among them, RS-Camera was used in the first six experiments, and NV-Camera was used in the last two experiments. All the experiments were conducted in the room, as shown in Fig. R4(a). The first experiment tested a scene with a natural light intensity of 290.0 lux, and the second to the sixth experiments used an adjustable light source, as shown in Fig. R4(b-f). From the results of RS-Camera, we can see that when the light intensity is greater than or equal to 5.9 lux, the object detection algorithm can complete the target tracking task. It was observed in experiment that when the light intensity is around 2.7 lux, the performance of the object detection algorithm of RS-camera is less effective, and it is difficult to meet the stable detection. Fig. R4 (f) shows that when the light intensity is lower (2.7 lux), the detection cannot be completed, and in some areas we cannot detect the target. Therefore, it

can be concluded that when the light intensity is less than 5.9 lux or it is completely black, the real-time tracking function cannot be effectively realized.

Next, we changed the responsibility for obtaining the target position information to the NV-Camera, and carried out the last two experiments, as shown in Fig. R4(g-h). The light intensity from left to right was 5.9 lux and 0.0 lux, respectively, and 0.0 lux represented completely dark. From the experimental results, it is observed that the NV-Camera can complete the detection of moving targets with an average confidence coefficient of over 95% under very low light intensity or even in completely dark. We denote that the NV-Camera and RS-Camera are both used as vision sensors, in which an image processing device reads image data from the camera and performs vision algorithm processing. The main difference between them is that NV-Camera can obtain the infrared image of the target in a dark environment, but without the depth of the target. In other words, the NV-Camera can only obtain the elevation and azimuth angles of the object in the camera's coordinate system. NV-Camera can automatically switch to night vision when the illumination is low or completely dark. In contrast, the RS-Camera can get the depth of the target in a well-lit environment so as to obtain the 3D coordinates of the target, but is not able to obtain effective images in a dark environment. For our system, NV-Camera is used in conjunction with the RS-Camera. When the illumination is below 5.9 lux, it is switched to NV-Camera to complete the system operation. Under good lighting conditions, RS-Camera is used to obtain the detailed position of the object. (see Supplementary Movie 5 for details).

Fig. R4 Performance of the object detection algorithm under different light intensities. Here, (a-h), indoor, we used adjustable light source, and carried out experiments with different light intensity. **(First and second rows, RS-camera, left to right: 290.0 lux, 102.8 lux, 30.4 lux, 9.6 lux, 5.9 lux and 2.7 lux. The third row, NV-camera, left to right: 5.9 lux and 0.0 lux).** Blue foam was put on the floor to control the reflection of the

floor. When the light intensity is low (5.9 lux) or completely dark (0 lux), the NV-Camera helps to obtain the infrared image of the target. The NV-Camera cannot get the depth of the target, so only the elevation and azimuth angles of the object are given in (i).

In the revised manuscript, we added a comment to the discussion in subsection “Moving target detection and identification” as:

“The appropriate upgrades to the hardware in the system are good for more complex scenarios, such as the infrared thermal imagers which also have important applications in industry and temperature monitoring. Therefore, a night version infrared-cut camera (NV-Camera) as an aid to solve the detection task under the condition of limited ambient light or completely dark. Experimental results under different light intensities, demonstrate that when the light intensity is low, the system can switch from the RS-Camera to NV-Camera to complete the target detection task (see Supplementary Note 13 and Supplementary Movie 5 for details).”

We added a new Supplementary Note 13 to illustrate the feasibility of adding night-vision cameras to complete the detection task.

We also added a new Supplementary Movie 5 to present the experiment.

Referee #1 -- Comment 3:

3. The experiments were conducted in an anechoic chamber, which is considered an ideal lab environment. However, in realistic environments, including indoor and dense urban outdoor scenarios, both the target tracking as well as communication might not be as simple as shown in the results. For example, there might be multiple similar objects present with the target, or the target might be temporarily blocked by other objects. In this case, how would the tracking algorithm perform? It would be better if the authors could discuss this issue.

Authors Response:

Thank you very much for this professional comment. Multi-object tracking (MOT) is an important problem in computer vision. MOT aims at estimating bounding boxes and identities of objects in progress. Currently, tracking-by-detection is the most effective paradigm for MOT¹.

Similar target interference and **target occlusion** are two key problems for MOT. To solve the problem of **similar target interference**, multiple targets in the visual field are numbered and the corresponding numbers of them in the video stream are guaranteed to remain unchanged. This task can be completed by matching between the results of object detection in the preceding frame and the following one, where the two detected boxes with high similarity are considered to be the same target and assigned the same number. In the deep simple online and realtime tracking (SORT) algorithm proposed in literature², the similarity consists of the appearance feature similarity, which is the cosine distance between the features extracted by the convolutional neural network, and the spatial information similarity, which is

the mahalanobis distance between the two detection boxes. The cost matrix between the tracker and the current detection box of frame is obtained by calculating the similarity, and then the Hungarian algorithm is used to find the optimal match. For the problem of **target occlusion**, the deep SORT algorithm introduces a Kalman filter to solve the problem of transient target occlusion and the problem of missing individual frame detection. The algorithm initializes a Kalman filter for each tracker, and after the optimal match is obtained by the Hungarian algorithm, there are three kind state for trackers and detection boxes at this moment: successfully matched trackers and detection boxes, unmatched trackers, and unmatched detection boxes. (1) for the successfully matched tracker and detection boxes, the detection box is updated as a new observation to the Kalman filter; (2) for the unmatched tracker, the predicted value of the Kalman filter is used as the target state of the frame; (3) for the unmatched detection box, it is initialized as a new tracker and assigned a new number. In addition, the algorithm sets the maximum survival time of the Kalman filter, i.e. when no observation is obtained for n consecutive frames of the tracker, the target is considered lost and the information of that number is cleared.

In order to verify the performance of the proposed system when multiple similar targets and target occlusion are included, we conducted **three groups of experiments**. The first group is in the scenario when multiple similar targets are included, and the second one group is when the target is temporarily occluded. We demonstrate that the algorithm runs stable enough to complete detection and tracking tasks in these two scenarios. And the third group is the verification of energy reception in the case of multiple similar targets.

In the first group of experiments, we numbered multiple targets, and then manually entered the number of the tracked target. Location of the tracked target is presented in black. As shown in Fig. R5 (a), location of car No. 1 was firstly tracked as the cars moving from right to left, and then the tracked target was switched to No. 3 (see Fig. R5 (b)), and then to No. 2 (see Fig. R5 (c)). Fig. R5 (d) shows the information processed for all targets. From top to bottom, the confidence degree of car recognition, the location information of each car, and the number of the tracked car were respectively recorded. From the experimental results, it is verified that each car can be switched at any time, and the algorithm runs stable enough to complete the detection and tracking tasks with multiple similar targets.

In the second group of experiments, the target was temporarily occluded by the blue foam of shelter. Fig. R5 (e-g) respectively shows the recognition of the target before occlusion, the recognition when the target is occluded, and the reappearance of the target after occlusion. Fig. R5 (h) displays the information of the object detection when the target was occluded. From the experimental results, it can be seen that the tracking box of the car keeps moving even when the car is occluded by the shelter. It is judged and predicted by the situation of previous frame rates. In view of this, we conclude that the algorithm runs stably and can complete the tracking task when the target is occluded.

Fig. R5 Performance of the object detection algorithm in scenarios where (a-d) multiple similar targets and (e-h) temporary target occlusion happened. Here, two red and one blue cars of the same model were used as multiple similar targets. (a-c) are three cars moving from the right side to the left side of the scene. In the movement, the three targets are individually numbered and the position information of the tracked one is presented in black. (e-h) A moving target is temporarily blocked by the blue foam of shelter, but the tracking box can predict the target (as indicated in (f)) and thus complete the tracking task (as given in figure (h)). (see Supplementary Movie 2 for details).

In the third experiment, we rely on a prototype of the DPM to demonstrate that the tracked target can effectively receive energy from the source when multiple similar targets are involved. The experimental setup is the same as the one in the section of “**Moving target detection and identification**” of the original submission. The DPM as the transmitter was fed with a linearly polarized horn antenna connected to a vector network analyzer (VNA), and a patch antenna designed at 5.77 GHz as the receiver was located in the middle of the moving path of the cars. The VNA tested the energy received by the patch antenna (in term of S_{21}) to verify the multi-object detection. Fig. R6 (a-d) record four typical states in the movement of the cars. The blue car (numbered 1) ran first from right to left, but was not selected as the tracked target, and hence the energy received by the patch remained basically unchanged and low, as is indicated by the S_{21} curve in (a). Then the red car (numbered 3), which was chosen as the tracked target, started to run. As it moved from right to left, the S_{21} curve increased in (c) when it was near the patch and decreased in (d) when it left the patch. The other red car (numbered 2) was at rest as the reference of multiple targets. Fig. R6 (e) is the flow chart of the movements in the experiment and (f) shows the S_{21} measured in VNA over time. Clearly, this system can effectively fulfill communication to the tracked target even when similar targets exist nearby.

Fig. R6 Experiments to verify the tracking scheme when multiple similar targets are involved. Two red cars and one blue car of the same model were used as the multiple similar targets. We put the patch receiver in the middle of the moving path of the cars, and the VNA tested the received energy in term of S21 to verify the multi-object detection. (a-d) record four typical states in the movement of the cars. The blue car (numbered 1) ran first from right to left, but was not selected as the tracked target, and the energy received by the patch remained basically unchanged and low, as is indicated by the S21 curve in (a). Then the red car (numbered 3), which was chosen as the tracked target, started to run. As it moved from right to left, the S21 curve increased in (c) when it was near the patch and decreased again in (d) when it left the patch. The other red car (numbered 2) was at rest as the reference of multiple targets. (f) shows the S21 result of VNA over time. (e) is the flow chart of the movements in the experiment. (see Supplementary Movie 3 for details).

1. Zhang, Y., Wang, C., Wang, X., Zeng, W. & Liu, W. FairMOT: On the Fairness of Detection and Re-identification in Multiple Object Tracking. *International Journal of Computer Vision* **129**, 3069–3087 (2021).
2. Wojke, N., Bewley, A. & Paulus, D. Simple online and realtime tracking with a deep association metric. in *2017 IEEE International Conference on Image Processing (ICIP)* 3645–3649 (2017).

In the new submission, we added a comment to the discussion in subsection “Moving target detection and identification” of our main text:

“In fact, multiple similar targets and target occlusion are two key problems in the field of target tracking. To solve the problem of similar target interference in the visual field, multiple targets are firstly numbered and the corresponding number of them in the video stream is guaranteed to remain unchanged. And the deep SORT algorithm introduces the Kalman filter to solve the problem of transient target occlusion and the problem of missing individual frame detection. The performance of object detection algorithm based on the RS-Camera in above scenarios is presented in Supplementary Note 11 and Supplementary Movies 2 and 3. When multiple targets with different characteristics exist, the YOLOv4-tiny target detection algorithm can classify the targets in the field of vision at the same time, and decide the categories that the targets belong to. By judging the category, the position information of specified target is extracted, and the beam is controlled to point to the specifically tracked target (see Supplementary Note 12 and Supplementary Movie 4 for details).”

We added a new Supplementary Note 11 to illustrate the performance of the algorithm in the case of multiple similar targets and the target might be temporarily blocked.

We also added a new Supplementary Movies 2 and 3 to present the experiment.

In the original manuscript, the description of the experiment with two different targets moves to a new Supplementary Note 12.

Referee #1 -- Comment 4:

4. With respect to communications in a more practical environment, the interference at 5.8 GHz in realistic environments could be high, since this is an ISM band. How does the designed DPM solve the potential issue of interference from other actively communicating devices operating at a similar frequency?

Authors Response:

We thank the reviewer for this constructive comment. Wireless communication in intelligent tracking may be susceptible to strong EM interferences and the effectiveness of it can be seriously affected. Therefore, eliminating the similar-frequency EM interferences is vital for the recognition ability and action reliability of the intelligent system. For the proposed system, the programmable metasurface itself help to depress the EM interference from practical environment. The digital programmable metasurface (DPM) modulates incident waves from the feeding horn and creates flexible and controllable radiating beam towards the moving targets captured by the camera. To guarantee the communication performance, interferences from other actively communicating devices operating at a similar frequency or in the same operating band should be effectively depressed.

Firstly, we demonstrate that the DPM is able to solve the potential issue of interference from other actively communicating devices operating **at a similar frequency**. The reflected phase responses of the coding element and the calculated 2D far-field patterns are plotted in Fig. R7 (a-c). When the phase difference between the two states (“00” and “10”) of the element is $180^\circ \pm 37^\circ$, the far-field patterns at 5.7 to 5.9 GHz are quite good except for some

acceptable increase of sidelobes and the center frequency is 5.8 GHz. In contrast, at a similar frequency outside the operating band, e.g., at 5.6 GHz and 6.0 GHz, the DPM can no longer create directive beam, as is shown in Fig. R7(c). In view of this, we conclude that other actively communicating devices operating outside the band of 5.7-5.9 GHz cannot disturb the communication because they only result in very weak reflected EM energy towards the target.

Secondly, we demonstrate that the DPM can also depress **in-band interference** from devices other than the feeding source. The coding pattern of the DPM should be designed with regard to the position of the feeding source, so as to complete phase compensation and achieve efficient deflection towards the direction of the target. For example, Fig. R7 (d-f) give the coding patterns of 20° deflection when the feeding horn is at different positions. In figure (d) and (f), the horn is 300 mm away from the DPM and the incident angles (in terms of θ and φ) are $(0^\circ, 0^\circ)$ and $(45^\circ, 0^\circ)$ respectively. In figure (e), the horn is 600 mm away from the DPM, and the incident angle is $(0^\circ, 0^\circ)$. Blue patches denote OFF-state PIN diodes, and yellow patches denote ON-state PIN diodes. Clearly, the coding sequence of DPM is determined by the position of its feeding source and the required beam deflection. The same coding sequence will create different beam deflection for the external sources at different locations, even though the sources are in the operating frequency band of the DPM. Therefore, interference of external sources can be effectively depressed by the DPM itself.

Fig. R7. (a) The reflected phase responses of the coding element when the PIN diodes are switched ON and OFF in case of y-polarization. (b) Calculated 2D far-field results at 5.7 GHz, 5.8 GHz, and 5.9 GHz. (c) Calculated 2D far-field results at 5.6 GHz, and 6.0 GHz. (d-f) Digital coding schemes for beams towards 20° in the E-plane. Blue patches denote the OFF-state PIN diodes and yellow patches denote the ON-state ones. Different incident angles and distances correspond to different DPM coding patterns. In (d), the feeding horn is 300 mm away from the metasurface and the incident angle (θ , φ) is $(0, 0)$. In (e) the horn is 600 mm away and the incident angle is $(0, 0)$. In (f) it is 300 mm away and the incident angle is $(45^\circ, 0)$.

Next, we investigate the influence of interference when its power increases. In the commercial software CST, we established a rectangular waveguide as the interference source, named Port 2 as shown in Fig. R8(a). Port 2 is about 600 mm away from the DPM with an incident angle of 15° . The feeding horn of DPM is 300 mm away and is named Port 1. The deflection angle (θ, φ) corresponding to the encoded pattern of DPM is $(24^\circ, 0^\circ)$. We carried out six cases of simulation, in which the amplitudes of Port 1 and Port 2 are 1:0, 0:1, 1:1, 1:0.5, and 1:0.1 respectively, the last case is when the amplitudes of port 1 and port 2 are 1:0.1, the phase difference is 180 degrees, as given in Fig. R8(b). From Fig. R8(c, d) to observe the influence of interference sources on the beam in more detail. It can be seen from the results that when the energy of the interference source is low, for example, when the energy of the interference is one tenth of that of the feed, the influence on the main beam is neglectable. However, when the energy of the interference is much higher than that of the feed, the beam width and directivity are affected to some extent, and the sidelobes are significantly higher. Therefore, the DPM can depress in-band interference with comparable power. And when the power of external interference is relatively high, we need to increase the feeding power of the DPM.

Fig. R8. (a) Simulation model of the proposed DPM with two-port excitation. (b) A description of the different excitation amplitudes of the two ports. (c, d) the co-simulation far-field results under different excitations.

Moreover, dual-polarized DPM can help to depress the interference. In a dual-polarized DPM, each element can independently modulate the phases for different polarizations and reflect dual-polarized signals with high isolation. Each polarization is controlled in real time through an individual interface to FPGA. For the interference at a similar frequency, DPM is used to regulate the signals of the two polarization channels, and the better polarization channel is selected by observing the bit error rate received by the signals at the receiver. For the verification of dual-polarization performance, please refer to the answer to Referee #1 -- Comment 5 for details.

Below is the discussion of possible methods for the same frequency interference elimination. Similar frequency interference is a problem in many scenarios such as weather radar, distorting radar variable estimation, etc. Several methods have been adopted and studied by many scholars for solving the interference problem. For example: 1. setting an isolation board. 2. Design specific property of the transmitter or the receiver to eliminate interference. In addition, some filters are often adopted for solving interference problems. For example, adaptive notch filter¹, object-orientated spectral polarimetric (OBSPol) filter², and nonlinear filtering³. Sidelobe blanking⁴ is also used in communication systems to mitigate interference.

In ref⁵ and Fig. R9, the wireless sensing system exploits the antenna pattern diversity and software programmability of the dynamic metasurface antennas (DMA) to achieve high-performance wireless sensing. A general framework for DMA-based wireless sensing, and demonstrate the feasibility and benefits of the DMA in sensing using custom hardware. A general deep learning framework for RF sensing in the IoT has been proposed⁶. A potential solution is to use the antenna pattern diversity of DMA to generate rich high-dimensional channel measurements, and simulate the influence of environmental dynamics on the received signal. In other words, for a given sensing application, people can learn a set of common features shared between different DMA antenna patterns. Due to the change of antenna pattern or environment, the learned features should be robust to signal diversity. Therefore, the receiver extracts the information of the interference signal through signal processing, and then feedback it to the reconfigurable metasurface to achieve an optimize coding pattern, so as to eliminate the interference of similar frequency.

Fig. R9. The end-to-end design of the DMA-based wireless sensing system⁵.

1. Chen, G., Ren, Z., Li, Y. & Zhang, T. A Method of Same Frequency Interference Elimination Based on Adaptive Notch Filter. in *2009 International Workshop on Intelligent Systems and Applications* 1–4 (2009).
2. Yin, J., Hoogeboom, P., Unal, C. & Russchenberg, H. Radio Frequency Interference Characterization and Mitigation for Polarimetric Weather Radar: A Study Case. *IEEE Transactions on Geoscience and Remote Sensing* **60**, 1–16 (2022).
3. Arce, G. R. & Hasan, S. R. Interference term elimination of the discrete wigner distribution using nonlinear filtering. in *2000 10th European Signal Processing Conference* 1–3 (2000).
4. Shnidman, D. A. & Shnidman, N. R. Sidelobe Blanking with Expanded Models. *IEEE Transactions on Aerospace and Electronic Systems* **47**, 790–805 (2011).
5. Lan, G. *et al.* Wireless Sensing Using Dynamic Metasurface Antennas: Challenges and Opportunities. *IEEE Communications Magazine* **58**, 66–71 (2020).
6. Wang, X., Wang, X. & Mao, S. RF Sensing in the Internet of Things: A General Deep Learning Framework. *IEEE Communications Magazine* **56**, 62–67 (2018).

In the new submission, we added a comment to the discussion in subsection “RF signal detection” of our main text:

“With respect to communications in a more practical environment, the interference at 5.8 GHz in realistic environments could be high, and the designed DPM can solve the potential interference by its own characteristics and attached sensing devices. More discussions are illustrated in Supplementary Note 17.”

We added a comment to the discussion in subsection “Real-time wireless transmissions” of our main text:

“Moreover, in a more practical environment, there will be interference problems such as other communication devices in adjacent frequency bands. The programmable ability and dual-polarization performance of DPM itself, help to eliminate the interference together with wireless sensing and other potential ways (see Supplementary Note 17 for details).”

We also added a new Supplementary Note 17 on contents of our response letter.

Referee #1 -- Comment 5:

5. In the DPM design, what is the role of dual polarization in experiments? The experiments seem not to take advantage of the polarization diversity. It would be better if the authors could also discuss this matter.

Authors Response:

We apologize for the lack of explanation of the dual polarization function of DPM in the original manuscript, and we are glad that the referee gave us the opportunity to supplement the relevant information.

In the proposed DPM every element can reflect dual polarized EM waves and independently manipulate the phase of reflected waves of different polarizations with high isolation. Dual polarizations are controlled in real time through an individual interface to FPGA. The platform relies on time-modulated polarization switches and, by varying the duty cycle and time delays of the polarization channels, we can arbitrarily rotate the polarization at the operating frequency.

On the one hand, these features make it possible and efficient to utilize an additional degree of freedom, the polarization, to **improve the capacity and integration of the DPM-based communication system**. On the other hand, dual-polarized DPM can help to **depress the interference**. In a dual-polarized DPM, each element can independently modulate the phases for different polarizations and reflect dual-polarized signals with high isolation. For in-band interferences, DPM can regulate the signals in two polarization channels, and the better polarization channel is selected by observing the bit error rate received by the signals at the receiver.

To demonstrate the dual polarization performance of the proposed DPM, we added the following **two experiments**. Firstly, since we only presented the beam steering function of the DPM under y -polarization in the original manuscript, here we added the measured steerable beam on the E-plane from -40° to 40° with an increment of 10° under x -polarization, as given in Fig. R10. Secondly, we added an experiment of RF signal detection in the outdoor environment to verify the tracking performance of the dual-polarized system, as shown in Fig. R11. The following are the supplementary notes, experimental results and discussions.

In the first experiment, we tested the performance of beam steering when **x -polarized** feeding horn is adopted. We denote that this is the supplementary experiment to the one presented in the section of “**Beam steering by DPM**” in the original manuscript where y -polarized feeding horn was adopted. The experimental setup is shown in Fig. R10 (a, b). The DPM presents great performance of dynamic beam scanning controlled by the FPGA, and Fig. R10 (c) plots the measured beams on the E-plane from -40° to 40° with an increment of 10° . With the increment of the scanning angle, the gain decreases from 18.77 dB to 14.43 dB and the beam width becomes wider due to the fact that the effective aperture of the DPMs becomes smaller as the scanning angle increases. Nevertheless, the good performance of designable radiation patterns and steerable power distributions guarantees the feasibility of the proposed intelligent tracking system in both polarizations.

Fig. R10. An x -polarized rectangular horn antenna is used to illuminate the DPM. (a) The far-field experimental setup in an anechoic chamber. (b) Front view of the fabricated DPM and the experimental setup. (c) The measured far-field patterns when beams on the E-plane vary from -40° to 40° at 5.8 GHz. The experiment verifies that for x -polarized incidence, the DPMs can also shape the far-field patterns in the spatial domain with the steerable beam.

In the second experiment, we carried out experiments in an outdoor environment to conduct the real-time tracking scheme in dual polarization. The testing sites were chosen at the campus of Southeast University (SEU). Fig. R11 shows the outdoor environment and the experimental setup, including the feeding horn, the DPM, the RF signal detector on the model car, the signal generator, and the FPGA module and control system. The RF signal detector was always placed on the moving car to detect the energy of the EM wave in real time. It consisted of a receiving patch antenna, a battery, a detector AD8317, and a microcontroller unit (MCU). Detector AD8317 was adopted to accurately measure the RF signal power in the band of 1MHz-10GHz and convert the RF input signal to the corresponding dB scale with accurate logarithmic consistency. The input of the detector AD8317 was connected to the receiving patch antenna, and the output of it was connected to MCU for monitoring and processing in real time. In this way, the portable detector without additional voltage source was realized.

We executed RF signal detection under dual-polarization wireless transmission channel in this experiment. Polarization state of the feeding horn and the receiving patch antenna were changed to perform different polarizations. In this realistic outdoor environment, the RS-camera can still correctly capture the moving target, which is the detector-loaded car here, in the identification process. Four curves in Fig. R11(d, e) respectively plot the voltage values obtained by the detector and the corresponding dB calibration values under dual-polarization.

When the detector and the car move together, the received energy is relatively stable with a high value. This number is well aligned with the power gain observed in the indoor test.

Fig. R11. Outdoor experiment of the intelligent tracking system. (a) Setup of the RF signal detection experiment. (b) y-polarized wireless transmission channel, (c) x-polarized wireless transmission channel. For scenarios with different polarizations, we changed the feeding polarization of the DPM and the orientation of the RF signal detection module attached to the car. RF signal changes under (d) y-polarization and (e) x-polarization when the detector moves with the car. The horizontal ordinate is the moving path of the car from the beginning to the end.

In the new submission, we added a comment to the discussion in subsection “Beam steering by DPM” of our main text:

“We note that the element itself is symmetrical and the performance of beam steering under the x-polarization is good (see Supplementary Note 4 for details).”

We added a comment to the discussion in subsection “RF signal detection” of our main text:

“Then, we carry out experiments in outdoor environment to conduct the real-time tracking scheme in dual polarizations. The testing sites are chosen at the campus of Southeast University. Fig. 6(c) shows the outdoor environment and experimental setup, including the transmitting horn, RF signal detector, DPM locations, signal generator, FPGA module and control system. This field trial is conducted outside the side entrance of our laboratory. The portable RF signal detector is attached to the car to observe the change of RF signals during the movement. Four curves in Fig. 6(d) plot the voltage values obtained by the detector and the corresponding dB calibration values under dual-polarizations, respectively. When the detector and the car move together, the received energy is relatively stable with high value. This number is well aligned with the power gain observed in the indoor test.”

We changed some images in subsection “RF signal detection” of Fig. 6 of our main text:

The main contents are Fig. 6(c) and Fig. 6(d). “(c) Outdoor experimental setup of the intelligent metasurface system. (d) The received RF signals under the y-polarization and x-polarization when the detector moves with the car, in which the horizontal ordinate is the moving path of the car from the beginning to the end.”

We also added a new Supplementary Note 4 to show the measured far-field results of the x-polarization, and added a new Supplementary Note 15 to show the detailed of outdoor test results.

Referee #1 -- Comment 6:

6. In the 2D vector that serves as the input to the ANN, what is the precision of the angles? Are they rounded to the nearest integer values to follow the 10-degree increment of the beams from the DPM?

Authors Response:

We appreciate very much this comment. In our designed network, precisions of both the elevation and azimuth angles can be as small as 1° , and so is the increment of the beams from the DPM. However, in the practical scenario of small-scale communication with a few users, the increment of 3° can meet the actual demand. So we set the increment of θ and φ to 3° for demonstration, and the detected angle was rounded to the nearest integer values (e.g., -3° , 0° , 3° , etc.)

In the new submission, we added a comment to the discussion in subsection “Experimental setup and environment” of our main text:

“In our designed network, the precisions of both elevation and azimuth angles can be as small as 1° . In the practical scenario of small-scale communications with a few users, the increment of θ and φ is set to 3° for demonstration, and the detected angle was rounded to the nearest integer value.”

Referee #1 -- Comment 7:

7. In Line 272 under subsection “Experimental setup and environment”, it is not clear what the term “every three times” means, if it is three seconds or three time samples. It would be better if the authors could further clarify this.

Authors Response:

We apologize for the possible confusion this sentence may cause. The point we want to convey is that, the RS-camera samples the object in each sampling frame, but the sampled data are returned to the network for processing and sent to FPGA for updating the voltage sequence after running three sampling frames. As shown in Fig. R12 below, we screenshot the three sampling frames during the running of the object detection algorithm, and denote them with red, blue and yellow boxes respectively. The position information of the detected object is obtained in all three frames of sampling, but only in the last frame of the information, sent to the network for processing, then the corresponding voltage sequence is sent to the FPGA, as marked in the yellow box.

Fig. R12. The screenshot image of the object detection algorithm in the process of running three frames. The position information of the detection object is obtained in all three frames of sampling, but only in the last frame of the information, sent to the network for processing, then the corresponding voltage sequence is sent to the FPGA, as marked in the yellow box.

In the new submission, we added a comment to the discussion in subsection “Experimental setup and environment” of our main text:

“but the RS-camera samples the targets every three frames and returns the sampled data only once to the designed ANN. Then the position information and the output voltage sequence are updated and sent to FPGA.”

Referee #1 -- Comment 8:

8. *In the subsection of “RF signal detection”, how are the voltage values measured by the detector converted to the power values? It would be better for the authors to present more details.*

Authors Response:

Thank you very much for this professional question. The AD8317 is a 6-stage demodulating logarithmic amplifier, specifically designed for the use in RF measurement and power control applications at frequencies up to 10 GHz¹. A block diagram and basic connections are shown in Fig. R13(a, b). Using precision biasing, the gain is stabilized over temperature and supply variations. The overall dc gain is high, due to the cascaded nature of the gain stages. An offset compensation loop is included to correct for offsets within the cascaded cells. At the output of each of the gain stages, a square-law detector cell is used to rectify the signal. The RF signal voltages are converted to a fluctuating differential current having an average value that increases with signal level. Along with the six gain stages and detector cells, an additional detector is included at the input of the AD8317, providing 50 dB dynamic range in total.

The output voltage vs. input signal voltage of the AD8317 is linear-in-dB over a multidecade range. The equation for this function is

$$\begin{aligned} V_{OUT} &= X \times V_{SLOPE/DEC} \times \log_{10}(V_{IN}/ V_{INTERCEPT}) \\ &= X \times V_{SLOPE/20dB} \times 20 \times \log_{10}(V_{IN}/ V_{INTERCEPT}) \end{aligned}$$

where:

X is the feedback factor in $V_{SET} = V_{OUT}/X$.

$V_{SLOPE/DEC}$ is nominally -440 mV/decade, or -22 mV/dB.

$V_{INTERCEPT}$ is the x-axis intercept of the linear-in-dB portion of the V_{OUT} vs. P_{IN} curve (see Fig. R13(d)). $V_{SLOPE/DEC}$ represents the volts/decade. A decade corresponds to 20 dB; $V_{SLOPE/DEC}/20 = V_{SLOPE/20dB}$ represents the slope in volts/dB.

These parameters are very stable against supply and temperature variations. The input dynamic range is typically 55 dB (referenced to 50 Ω) with less than ± 3 dB error. The AD8317 has 6 ns/10 ns response time (fall time/rise time) that enables RF burst detection to a pulse rate

of beyond 50 MHz. The device provides unprecedented logarithmic intercept stability vs. ambient temperature conditions. A supply of 3.0 V to 5.5 V is required to power the device. Fig. R13 (d, e) is results of V_{OUT} and input amplitude P_{IN} at 5.8 GHz based on datasheet and actual measurement, and we take the actual result as reference.

Fig. R13 (a) The detector AD8317 is connected to the patch antenna from the front. On the back of detector, the battery supplies powers to MCU, and the power port on MCU supplies powers to AD8317. (b) Functional block diagram of AD8317. (c) Basic Connections of AD8317. (d) V_{OUT} and log conformance vs. input amplitude P_{IN} at 5.8 GHz. Typical performance characteristics in the datasheet of AD8317. (e) V_{OUT} vs. input amplitude P_{IN} at 5.8 GHz, the measured results.

1. Analog Devices. AD8317:1MHz to 10GHz, 55dB Log Detector/Controller Data Sheet. (2008). <https://www.analog.com/cn/products/ad8317.html>

We changed some images of Fig. 6 in subsection “RF signal detection” of our main text:

“The detector AD8317 is connected to the patch antenna from the front. On the back of detector, the battery supplies powers to MCU, and the power port on MCU supplies powers to AD8317. (b) Functional block diagram of AD8317.” and the picture in the original manuscript, this part of the content is moved to the Supplementary Note 14.

We also added a new **Supplementary Note 14** to show the working mechanism of the detector AD8317.

Referee #1 -- Comment 9:

9. In the subsection of “Real-time wireless transmissions”, the bit error rate performance needs quantitative analysis, in addition to the photos shown in Fig. 7, to help readers with a better understanding.

Authors Response:

Bit error rate (BER) is an important index in wireless communication, which can express the accuracy of data transmission. We are very grateful for the reviewer’s valuable suggestions and the opportunity to improve the experiment.

To test the BER value, a realistic wireless communication system was built to perform experiments of direct data transmission in an indoor scenario, as shown in Fig. R14(a) and R15(a). The experimental setup is applied to test the BER performance of the wireless communication system and the compatibility of the wireless communication system with different modulation modes. The vector signal transceiver (VST, PXIe-5841, National Instruments Corp.) in the figure is used for the BER measurement. The instrument has the function of setting different operating frequencies, modulation modes and bit transmission rates. We take the DPM fed by a linearly polarized horn antenna as the transmitting terminal, and the receiving antenna is fixed somewhere on the moving path of the target (in Fig. R14(a)) or on the moving model car (in Fig. R15(a)). The transmitter and the receiver are kept the same height from the ground. The transmitter is connected to the signal output port of the VST, and the receiver is connected to the signal input port of the VST.

The commonly used modulation modes include quadrature phase shift keying (QPSK), quadrature amplitude modulation (QAM), and so on. In this BER experiment, the sinusoidal carrier of QPSK signal has four possible discrete phase states, and each carrier phase carries two binary symbols. Bit rate, also known as “binary bit rate”, is used to describe the transmission rate of wireless communication systems. It represents the number of bits transmitted per unit time (commonly written as bps). Generally speaking, by changing the parameter settings of the VST, the signal transmission test with different bit rates can be realized.

Fig. R14 (a) Experimental setup of BER testing. Two colorful pictures were transmitted from the transmitter (the horn) to the standing receiver (the receiving antenna), with an information transmission rate of 170 Mbps at the frequency of 5.8 GHz. (b) The measured value of BER.

In the two experiments of BER test, we set the modulation mode to QPSK and the transmission rate to 170 Mbps, read the data in the display panel of the VST and record the results. In the first experiment, it is observed in Fig. R14(b) that when the moving car is close to the receiving antenna, the wireless communication is reliable and the BER is stable at 10^{-5} , whilst when the moving car is far away from the receiving antenna, the beam is no longer

made towards the receiving antenna and the BER value is very high. In contrast, in the second experiment, it is observed in Fig. R15(b) that the wireless communication is always reliable because the beam is always made towards the receiving antenna on the moving model car, and the BER value is stable at about 10^{-5} .

Fig. R15 (a) Experimental setup of BER testing when the receiving antenna is fixed on the moving car. (b) The measured value of BER.

We added a comment to the discussion in subsection “Real-time wireless transmissions” of our main text:

“Here, the vector signal transceiver (VST, PXIe-5841, National Instruments Corp.) is used for the bit error rate (BER) measurement, in which we set the modulation mode as QPSK and the transmission rate as 170 Mbps. When the channel is in the acceptance region, the BER is stable at 10^{-5} (see Supplementary Note 16).”

We also added a new **Supplementary Note 16** to exhibit the experimental setup and results of BER testing.

Referee #1 -- Comment 10:

10. The energy consumption of the proposed design, including object detection, tracking, and communication, is another aspect that might affect the system performance. Some numerical results would be better to understand this matter.

Authors Response:

We would like to thank the reviewer for this valuable suggestion. The supplement on energy consumption is very important to this work.

We categorize the devices involved in this design and list their power consumption in Table R2. The RS-camera and the control system (with a laptop included) are combined as a whole. We can monitor the power consumption on the laptop when the control system is in standby state (as shown in Fig. R16 (c)) and when it is running the RS-camera for tracking tasks and sending FPGA instructions (as shown in Fig. R16 (d)). The working state of DPM is mainly determined by the power supply of FPGA, so the power consumption of FPGA and DPM is displayed in one column. In the experiment of RF signal detection, the transmitter is a microwave signal generator (Keysight E8267D) and the detecting module is mainly composed of a detector AD8317 and a micro controller unit (MCU). In the experiment of real-time wireless transmissions, the image transmission module is responsible for most power consumption.

Considering that the theoretical calculation or self-displayed power may not be available in practice, we used a power detector DL333501 (as shown in Fig. R16(a)) to test the energy consumption of the design, as given in Table R2. DPM is powered by FPGA, so the total power consumption of the FPGA board and the pinboard in Fig. R16 (b) is 13.129 W. We denote that FPGA devices that consume less power can reduce this value even more. For the RS-camera and the control system, the displayed powers are 8 W and 27 W for the “standby” and “running” states, respectively. Since the camera is connected to the Laptop through the USB interface, no extra power supply is needed. The values are recorded in Table.R2.

Fig. R16 Energy consumption of the design. (a) The power detector DL333501 used to measure the power consumption of instruments in the design. (b) The measured power consumptions of the main instrument in use. (c, d) The power consumption displayed on the laptop when the control system is in “standby” state and “running” state. ON and OFF represent the “running” and “standby” states of the intelligent tracking algorithm, respectively.

Table.R2 Energy consumption of the design

Instruments	Implications	The value of the system
RS-camera and the control system	Standby state	8 W
	Running state	27 W
FPGA with DPM	DPM is powered by FPGA, so the sum of FPGA and Pinboard is the power consumption of this part	13.129 W
Signal generator	-10 dBm	Variable
Detector	A detector AD8317 and MCU	0.5×10^{-3} W
Real-time wireless transmissions	The image transmission module	5×10^{-3} W

Next, we take into consideration of the change of performance when the target is located at different distances from the DPM under different transmitting powers. We aligned the target in the directions of $\theta = -30^\circ$ and $\theta = 20^\circ$ and tested the received energy when it was 1.5 m and 3.0 m away from the DPM. Fig. R17 (a, b) shows the schematic of the testing scenario, where the signal generator (SG) is connected to the horn as the transmitting source and the receiving antenna is connected to the spectrum analyzer (SA) to measure the power of the received signal. We measured the received power values when the target was 1.5 m and 3.0 m away from DPM and the transmitting power was set to -10 dBm, 0 dBm, 10 dBm and 20 dBm respectively, at the two directions of -30° and 20° . Fig. R17 (c, d) plots the measured results. It is observed that with the same transmitting power, the received power decreases by an average of 2 dBm. As the transmitting power increases, the received power also increases by the same amount, as indicated by the curves in Fig. R17 (c, d). In view of this, we conclude that the

total energy consumption is also dependent on the power required by the receivers, the number of receivers, and the communication distance.

Some consideration could be involved in future work to further reduce the power consumption. We can use a single FPGA development board connected to the camera to run the target tracking algorithm and realize the task of sending voltage sequences to the DPM. In addition, the use of low power varactor and numerical control rheostat can also help to reduce the power consumption of the system.

Fig. R17. (a) Schematic of the experimental setup to measure the power of received signal. (b) The transmitting horn is connected to the signal generator (SG), and the receiving antenna is connected to the spectrum analyzer (SA). The power of received signal is plotted when the target is aligned in the directions of (c) -30° and (d) 20°.

We added a comment to the discussion in subsection “Real-time wireless transmissions” of our main text:

“We estimate the energy consumption of the proposed design in Supplementary Note 18, including the object detection, communication, and power supplies. The performance of the

design under different input powers is also discussed and tested, as presented in Supplementary Note 18.”

We also added a new Supplementary Note 18 to show the energy consumption of the proposed design.

General comments from Referee #2:

This paper presents the development of a system allowing an active reconfiguration of a metasurface whose radiation is controlled by a convolutional neural network. This work is conducted in an incremental framework based on physical solutions introduced in previous publications. This system is fed by optical data and facilitates beamforming towards a target in an ideal electromagnetic environment (anechoic chamber). The aimed applications are oriented towards target tracking and wireless communications, in a context motivated by the increasing development of reconfigurable means and allowing the possible increase of the transferred data rates.

Authors Response:

We are grateful to the referee for the detailed review and constructive comments, which greatly help us to improve the quality of the manuscript.

Specific comments from Referee #2:

Referee #2 -- Comment 1:

The bibliography is almost entirely oriented towards the work of the authors or their direct collaborators. The presentation of the numerous applications of these new technologies in the introductory part thus eludes in part or completely the contributions of the many research groups which however contributed to found some of the cited applications. Following an extraction of all the papers cited, the figure inserted in the following page presents a graph of interaction between authors. The latter allows to highlight the very high rate of self-citation of some of the contributors of the paper I have been charged to evaluate. Insofar as the positioning of this work appears almost absent in comparison with numerous pioneering contributions proposed by other research groups, the authors' work does not allow the reader who is not familiar with this field to form an objective opinion of the real contribution of this paper to the literature.

Fig. 1. Interaction graph of authors extracted from the bibliography of the evaluated paper, highlighting a very high rate of self-citation and the absence of contributions from active/pioneer groups among the applications studied in the introduction of this paper.

We appreciate very much to the referee for the valuable comments, and apologize for the lack of numerous pioneering contributions from active/pioneer groups in the original manuscript. We have added a series of important pioneering contributions in the Introduction of the revised manuscript, so as to provide the readers with a more comprehensive overview of the background. Please refer to the revised **Introduction** for details. All changes are highlighted in the revised manuscript.

Specific comments from Referee #2:

Referee #2 -- Comment 2:

The proposed demonstration is convincing but the complexity of the proposed approach in such a controlled environment does not seem justified. To summarize the proposed approach, the principle relies first on the optical detection of the angle of a target with respect to the electromagnetic surface reference frame. Beamforming is then performed from the latter using phase modulations coded on 1bit according to patterns that can be readily justified analytically. What is the added value of a technique based on the exploitation of artificial neural networks compared to solutions that are simpler to implement, potentially faster and where physical hindsight is not abandoned in favor of machine learning solutions? It is necessary that the authors can justify the usefulness of the proposed approach, especially in experimental conditions as controlled as in an anechoic chamber.

Authors Response:

We appreciate very much to the referee for these valuable comments, and apologize for the lack of detailed descriptions of the artificial neural networks presented in the first manuscript.

To your question (1), the added value of the presented artificial neural networks (ANN) is summarized from three points: a) compared to the theoretical calculation like back-projection (BP)^{1,2}, the presented algorithm has better **beam accuracy and sidelobe performance**; b) compared to the nonlinear optimization algorithms like genetic algorithm³ (GA) and particle swarm optimization⁴ (PSO), the presented method raises a much **faster speed** to obtain the target coding matrix; c) for realistic environment, the presented ANN is able to **overcome some interference** such as environmental multipath scattering and other interference sources. More details and analysis are provided in the following.

To your question (2), indeed, the controlled condition like an anechoic chamber cannot fully exhibit the advantages of the ANN. Therefore, we supplemented an **outdoor-field experiment** (please refer to the answer to Referee #1 -- Comment 5 for details) to demonstrate that the presented ANN can work in the complex environment. The measurement results proved that the ANN has the capabilities to solve the interference and guarantee the high signal-noise ratio (SNR) (please refer to the answer to Referee #1 -- Comment 9 for details).

In summary, our ANN method has the following advantages:

1. A higher speed. Respond in real time when coding the programmable metasurface. For a more intuitive comparison, we examine the computing time of the above three schemes to generate one coding matrix. It takes the back-projection approach 0.003 seconds, it takes the PSO algorithm about 25 seconds for a low sidelobe case in this paper. The ANN approach it takes an average of 0.002 seconds. The computation platform is Intel Core i7-9750H CPU @2.60GHz and accelerated by one Nvidia GTX 1650 GPU. Platforms are different, and different methods take different amounts of time, and while that's not a fair comparison, it still indicates that the ANN approach can provide not only accurate but also real-time responses when coding the programmable metasurface.
2. Performs well with complex scattering problems. For complex beam requirements, global results can be accurately output through relatively large datasets.
3. Anti-interference. The proposed ANN can work normally in the outdoor environment, showing good anti-interference characteristics. The digital programmable metasurface (DPM) modulates incident waves from the feeding horn and creates flexible and controllable radiating beams in dual polarization. The proposed ANN can be flexibly deployed in different directions of waves and specific usage scenarios. Please refer to the answer to Referee #1 -- Comment 4 and 5 for details. Compared with traditional computing methods, artificial neural networks can solve complex scattering field in real time. It also has the characteristics of considering the external EM environment and has stronger anti-interference ability. Assuming that the current EM environment and noise are stable, we can collect the EM environment information of the system in advance to customize an adaptive and easily deployed network in the current environment. In the case

of non-interference scenarios, the original parameters can be fine-tuned, without training from the beginning, and quickly deployed to various scenarios.

4. Although ANN usually requires a large time overhead in the training process, a well-trained ANN model has significant advantages in practical applications.

To specifically exhibit the above advantages, we compare the presented ANN with other methods in terms of speed, side-lobe and anti-interference. Table.R3 lists the speed and low-sidelobe performance after executions of BP, PSO optimization and the proposed ANN. The speed calculation of the three methods is based on the design of the 18×18 digital programmable metasurface (DPM). The low-sidelobe solution is not applicable (N/A) using the BP method, but is **available** through the PSO optimization and the proposed ANN method. For anti-interference capability, first of all we comprehensively consider low side-lobes⁸⁻¹⁰ which are important to communication SNR. In addition, the ANN is designed to use actual measurements as a training set to obtain more practical results when compared with the nonlinear optimization methods. The adjacent elements of the DPM are not independent, but a whole formed by interaction and coupling that are difficult to be optimized. Also, the higher order scattered waves are difficult to be predicted by the optimization methods. Therefore, from data collection to algorithm modeling to experimental demonstration, globally designed algorithms are required. In view of these, adopting the measured results as a training set to obtain more accurate output, ANN provides added value for **complex beam requirements and anti-interference problems**⁶.

Table.R3. The results derived from BP, PSO optimization, and the proposed ANN

Method	Speed ⁷	Low side lobe	Anti-interference ^{6,14}
Back-Projection	≤ 5 ms	N/A	Low
Nonlinear Optimization	≥ 20 s	Available	Adequate
ANN	≤ 5 ms	Available	High

More details about the comparisons and the presented ANN:

We will firstly briefly introduce the nonlinear optimization algorithm with low sidelobe and then introduce the details of artificial neural networks, in which **Methods A** shows particle swarm optimization approach for low sidelobe level (SLL) of the DPM; while **Methods B** is deep learning approach for the proposed intelligent tracking system.

A. Nonlinear Optimization Approach

The calculation of the metasurface coding matrix can be formulated as an optimization problem, to design a given scattering pattern of the metasurface. We generally use random nonlinear optimization algorithms, such as genetic algorithm³ (GA) or particle swarm

optimization⁴ (PSO), to approximate the designed optimal coding matrix through iteration. The nonlinear optimization algorithm uses randomness and other characteristics to find the global minimum, which is the coding matrix with the best performance of the metasurface. We use the PSO algorithm to optimize the beam. The following figure shows the coding matrices and simulated results for realizing a single beam at $(\theta, \varphi) = (25^\circ, 0^\circ)$, the SLL under BP method and PSO optimization is lower than -10.24 dB and -12.13 dB at 5.8 GHz, respectively. When PSO is used for low sidelobe optimization of DPM, the vector dimension is set to 100 and the maximum number of evolutionary iterations is 100.

Fig. R18 Single-beam coding matrices calculated by (a) back-projection and (b) PSO algorithm. In (c) shows the position where the element is flipped obtained by two approaches and (d) compares the beam pattern on the principle plane generated by these two approaches.

B. ANN Approach

The deep learning techniques combined with the metasurface can compute the coding matrices for complex beam patterns⁵⁻⁷. Regarding the design of reconfigurable metasurfaces at microwave frequencies, some scholars have proposed deep learning-assisted design schemes. According to those application scenarios, the network input of deep learning can be the radiation pattern⁷, spectrum information^{6,8,9}, or the information of incident waves¹⁰. Since the primary goal of the tracking system is to achieve beam alignment, we take the elevation and

azimuth angles of the scattering beam as the input of the network. The input consists of two angles of the scattering beam, as illustrated in Fig. R19. In this intelligent tracking system, the two angles (θ , φ) detected by the RS-camera are directly fed into the network for calculation. No additional operations are required to achieve specific network input forms.

The output of the artificial neural network is the coding matrix of the DPM, which can produce a single beam with low sidelobe that fulfills the realization of the input angle. Fig. R19 shows an example of the output. The output coding matrix consists of binary numbers “1” and “0”, which are corresponding to the ON and OFF states of the PIN diodes in the DPM. The reflected beam of the DPM can be manipulated through adjusting the coding pattern.

Fig. R19 Schematic of proposed artificial neural network (ANN).

Here, we give a detailed illustration of the proposed deep learning method of predicting coding matrix from θ and φ . Firstly, the input angles θ and φ are embedded as dense representations. The dimension of embedding is set to be 60. The representations of angles will then be concatenated and input into a 3-layer Multilayer perceptron (MLP). The dimension of each layer is also shown in the Fig. R19. We choose Rectified Linear Unit (ReLU) as the final activation function of the MLP and the dimension of the output from the MLP is 1282. The output will then be reshaped as square images with a side length of 128. The generated images based on the angles are inputs of the following ResNet34¹¹, which tries to predict the coding matrix. For each residual block, we implement batch normalization layer¹² and ReLU activation. We implement 16 residual blocks and every block have 2 convolution layers. With an additional convolution layer and the last fully-connected layer, we obtain 34 weighted layers in total. The dimension of the output is 18^2 so that it can be reshaped as the same size of the ground truth coding matrix. The activation function for output is sigmoid. We calculate the Binary Cross-Entropy / Log between the changed real patterns and those predicted ones as loss function. The formula of the loss is given as follows.

$$BCEloss = -\frac{1}{N} \sum_{i=1}^N ((truth(i) * \log(pred(i)) + (1 - truth(i)) * \log(1 - pred(i))))$$

where N is number of coding elements in a pattern, which equals to 18^2 here, $pred$ and $truth$ indicates the predicted and ground truth coding matrix, respectively. We utilize the Adam optimizer¹⁹ and the learning rate is set to be 2×10^{-5} . When we generate coding matrix with θ and φ in the test set, the positions in the output of ResNet34 with positive values will be encoded as “1” and those with negative values will be regarded as “0”. The prediction accuracy is given by the ratio of the correct elements in the predicted arrays. After proper numerical computation, the corresponding 2D scattering patterns will be obtained from the predicted coding arrays.

In the whole dataset, θ varies from -45° to 45° and φ varies from 0° to 360° . The value range of θ is not -90° to 90° (half-space), because the range of field angle that can be measured by the camera is only about 80° on the θ plane, and when the reflected beam of the DPM exceeds 70° , the main lobe of DPM is difficult to meet, and the phase difference between coding elements is not enough to achieve. Both θ and φ share the same step of variation of 1° , we randomly optimized low sidelobe codes from more than 12,000 angles, with 80% training, 20% testing. After 200 epochs of training, the prediction precision of coding positions on the test set is 95.32%, which suggests the effectiveness of our proposed model. The evolution of the loss during training is presented in Fig. R20. It can be seen that value of loss function declines rapidly as the parameters are optimized.

Fig. R20 The average loss and accuracy of the training and testing process

C. Discussion and conclusions:

In this work, we investigated the feasibility of applying deep learning techniques to encode the DPM for the single beam steering and the situation with low SLL.

The input of the network includes the parameters of the coding pattern in terms of θ and φ and the output is the coding matrix to program the DPM. This network is trained with data of coding matrices computed by back-projection or nonlinear optimization approaches. With the help of massive parallelization, the proposed deep network can compute the coding matrix in almost **real time** with **a great accuracy**. The results show that deep learning approaches can compute the coding matrices that generate almost identical beam patterns in milliseconds. We may modulate the EM wave with a good accuracy using DPM in both time and spatial domain simultaneously, with the help of efficient control circuits.

Moreover, the method based on deep learning can be improved by changing serial communication to parallel communication, leveraging hardware to do more computation, and providing a usable solution in real-time systems. It can solve complex scattering pattern in real time. In addition, when the environment changes, the network can be easily deployed in various environments after fine-tuning the network.

1. Yang, H. *et al.* A programmable metasurface with dynamic polarization, scattering and focusing control. *Scientific Reports* **6**, 35692 (2016).
2. Nayeri, P. & Elsherbeni, A. *Reflectarray Antennas: Theory, Designs, and Applications*. *Reflectarray Antennas: Theory, Designs, and Applications* (2018).
3. Donelli, M., Caorsi, S., Natale, F., Pastorino, M. & Massa, A. Linear Antenna Synthesis with a Hybrid Genetic Algorithm. *Prog Electromagn Res* **49**, (2004).
4. Robinson, J. & Rahmat-Samii, Y. Particle swarm optimization in electromagnetics. *IEEE Transactions on Antennas and Propagation* **52**, 397–407 (2004).
5. B. Sheen, J. Yang, X. Feng, & M. M. U. Chowdhury. A Deep Learning Based Modeling of Reconfigurable Intelligent Surface Assisted Wireless Communications for Phase Shift Configuration. *IEEE Open Journal of the Communications Society* **2**, 262–272 (2021).
6. Jia, Y. *et al.* In Situ Customized Illusion Enabled by Global Metasurface Reconstruction. *Advanced Functional Materials* **32**, (2022).
7. Shan, T., Pan, X., Li, M. & Xu, S. Coding Programmable Metasurfaces Based on Deep Learning Techniques. *IEEE Journal on Emerging and Selected Topics in Circuits and Systems* **PP**, 1–1 (2020).
8. Li, S., Liu, Z., Fu, S., Wang, Y. & Xu, F. Intelligent Beamforming via Physics-Inspired Neural Networks on Programmable Metasurface. *IEEE Transactions on Antennas and Propagation* **70**, 1–1 (2022).
9. Qian, C. *et al.* Deep-learning-enabled self-adaptive microwave cloak without human intervention. *Nature Photonics* **14**, 383–390 (2020).
10. He, K., Zhang, X., Ren, S. & Sun, J. Deep Residual Learning for Image Recognition. *CoRR* **abs/1512.03385**, (2015).
11. Ioffe, S. & Szegedy, C. Batch Normalization: Accelerating Deep Network Training by Reducing Internal Covariate Shift. (2015).

12. Kingma, D. & Ba, J. Adam: A Method for Stochastic Optimization. *International Conference on Learning Representations* (2014).

We added a comment to the discussion in subsection “Platforms of target detection” of our main text:

“In this study, ANN is designed to learn the coding matrix with low sidelobe characteristics obtained from the particle swarm optimization (PSO) methods, and the output has better beam accuracy with a much faster speed. Therefore, the proposed ANN has stronger abilities than the back-propagation method to solve complex scattering problems and faster speeds than the nonlinear optimization method. For realistic environment, the presented ANN has the advantages of lightweight, easy deployment and anti-interference (see Supplementary Note 8 for details).”

We also added a new Supplementary Note 7 to show structure and operating process of the pre-training ANN.

We added a new Supplementary Note 8 to show the advantages and usefulness of the proposed ANN approach.

We modified Fig. 1 and Fig. 4 in the revised manuscript.

General comments from Referee #3:

This manuscript presents an intelligent tracking system based on the digital programmable metasurface and computer vision. The authors use computer vision to locate the target to be tracked and send the position information to the intelligent metasurface system so as to realize self-adaptive beam tracking of the moving target. Experiments of radio frequency signal detection and wireless information transmission demonstrated the function of the proposed system and showed good stability. In my opinion, this work is interesting and the proposed intelligent tracking method can provide important and practical help for self-adaptive and smart communication systems. I can recommend this paper for publication after minor revision. Below are specific comments to be addressed.

Authors Response:

We are sincerely grateful for your positive comments on our work, and your professional and constructive suggestions are very helpful for us to improve the quality of the work. We have tried our best to revise the manuscript accordingly. Below are our point-by-point answers to all the questions and comments.

Specific comments from Referee #3:

Referee #3 -- Comment 1:

1. The authors put forward a 1-bit dual-polarized digital programmable metasurface, which seems to be controlled independently by each unit. However, the reconfigurable scheme of

units and configuration of feeding network are not clearly explained. Please give information on these issues, and also clarify the bandwidth of the reconfigurable metasurface.

Authors Response:

We thank the reviewer for this professional question. In the manuscript, we used dual-polarization elements to compose the metasurface. In the design of the elements, the feeding circuit needs to be compact with very limited influence on the performance of the metasurface. Fig. R21 (a) represents the outline of each metal layer of the element. The x -polarized feeder interconnection can be realized on third layer by changing the length of l_x . By changing the length of l_y , the unit can realize independent control of y -polarization in the fourth layer. Fig.R21 (b, c) show the working diagram of the feeder for each polarization. The reflected phase response of the element is given in Fig. R21 (d), showing that in the frequency band of 5.7-5.9 GHz the phase difference between the two states (“00” and “10”) of the element maintains within $180^\circ \pm 37^\circ$. The far-field pattern of the reflected beam with a specific coding matrix is calculated and plotted in Fig. R21 (e), demonstrating that for y -polarization the 20° beam is realized from 5.7 to 5.9 GHz with the center frequency being 5.8 GHz. The results for x -polarization are also presented in Fig. R7 (b) of this response letter. Thus, we conclude that the working bandwidth of the metasurface can be obtained to be about 200MHz.

Fig. R21. Perspective views of the coding element. (a) The outline of each metal layer in the element. (b, c) working diagram of the feeder for each polarization. (d) The reflected phase responses of the coding element when the PIN diodes are switched ON and OFF. (e) Calculated 2D far-field patterns at 5.7 GHz, 5.8 GHz and 5.9 GHz in y -polarization.

We added a new **Supplementary Note 4** on generalizations of our concepts, which includes the working diagram of the feeder for each polarization (Fig. R21(b, c)) and the working bandwidth of the metasurface is about 200MHz.

Referee #3 -- Comment 2:

2. In the intelligent tracking system, the RS-camera serves as the auxiliary to complete the task of moving target identification and tracking. What is the volume (width×depth×height) of the real scenario captured by the RS camera, and how does the system discriminates and switches between multiple targets?

Authors Response:

Thank you very much for this insightful comment. The actual captured range of the RS-camera is measured using the method of geometrical optics, as shown in Fig. R22(a). First, we compared the rectangular area captured by the RS-camera at the distance of $dx_1=25.7\text{ cm}$. The distance of dx_1 can be set according to the test platform. We put a white background board in the detection area of the camera, and then combined with the display area of the camera on the computer, we drew the boundary of the detection area, that is, the measured $lx_1=27.6\text{ cm}$ and $ly_1=19.2\text{ cm}$ on the $u-v$ plane. According to the datasheet, the RS-camera works ideally when the object is no farther than 3 m away from the camera. So, when $dx_1+dx_2=3\text{ m}$, we can deduce that lx_2 and ly_2 are 3.22 m and 2.24 m respectively. Therefore, we conclude that the volume of the real scenario taken by the RS-camera is a pyramidal area, whose apex is located at the position of the camera, height is 3 m and the bottom surface is a rectangle of $3.22\delta 2.24\text{ m}^2$, as indicated in Fig. R22(a). We calibrated the camera by processing the closer distance measurements in the manner shown in Fig. R22(a). Fig. R22(b) shows the actual measured scene displayed on the computer. When the target is about 20m away from the RS-Camera, the detection task can still be completed.

In the recognition algorithm of multiple targets, we can capture all targets, identify them and number all of the targets, and determine the number of the tracked target, and thus obtain the location information of the tracked target for processing. More details on this issue are presented in **the 3rd reply to Referee #1.**

Fig. R22. Mapping between the camera's view and the real view. (a) Captured range of the RS-camera based on the geometrical optics. (b) The actual measured scene displayed on the computer. Outdoor performance of the RS-Camera, when the target is about 20m away from the RS-Camera.

We have also added the contents of the response letter above to **Supplementary Note 1**, which includes method of measuring actual captured range and outdoor performance of the RS-Camera.

Referee #3 -- Comment 3:

3. On page 14, line 339, it is stated that "We collect the data sets for the detector-loaded car, so that the RS-camera can correctly capture the moving target in the identification process." The authors need to show the collection and processing of data sets.

Authors Response:

Thank you very much for this question. Details of the collection and processing of datasets were missing from the original manuscript. In this experiment, the RS-camera was

used to sample the tracked target (the model car with a portable RF signal detector attached on it here). The RS-camera takes pictures of the captured samples 3 times, and sends them as pictures for saving. We denote that the number of samples per second can be flexibly set, and in this work we chose the rate of 3 samples per second to ensure that there is no excessive repetition of samples.

During the sampling process, the moving target is captured at different positions of the field of view with different postures, so as to ensure that as many image samples of the target are collected as possible. Additional manual screening may help to remove some data with too much interference, leaving images with typical characteristics. These data are annotated through the tool “labelImg”, and the annotation information is saved as an xml file, which becomes the collection and processing of datasets, as shown in the Fig. R23 below. The sampling frequency and the label given to the target can be modified as required.

Fig. R23 Flow chart of dataset processing using labelImg

We have added the contents of the response letter above to Supplementary Note 11, which includes method of collection and production of data sets.

Referee #3 -- Comment 4:

4. I noticed that the SI provides the experiment on the response speed of the FPGA and the intelligent tracking system by logic analyzer. Is it possible to achieve higher frame rates using the methods presented in this article? Please clarify.

Authors Response:

Thank you for this inspiring comment. Speed of the system is an important issue for application. We re-use logic analyzer to conduct experiments on the response speed and intelligent tracking system of FPGA. We set the different sending frame rate of voltage sequences respectively. Among them, the detection frame rate of the RS-camera for the moving target is about 40 frames. We select the RS-camera to sample every three times and only send the corresponding voltage sequence to the FPGA once. The test result is 0.2059s as

shown in Fig. R24(b). You can also choose to send a voltage sequence to the FPGA with each sampling, which can increase the speed. Because the moving speed of the object is not very fast and in order to save energy consumption, we choose to send a voltage sequence to the FPGA every three frames detected. According to the situation of the existing system, if the detection of each frame is sent to the FPGA, the speed is about 79.22ms, and the test result is shown in Fig. R24(c). In addition, cameras with faster frame rates and computers with more processing speed can help our system to respond faster.

We added a comment to the discussion in Supplementary Note 10 “Description of switching speed of system” of our supplementary file:

“We can also choose to send a voltage sequence to the FPGA with each sampling, which can increase the speed. Because the moving speed of the object is not very fast and in order to save energy consumption, we choose to send a voltage sequence to the FPGA every three frames detected. According to the situation of the existing system, if the detection of each frame is sent to the FPGA, the speed is about 79.22ms. In addition, cameras with faster frame rates and computers with more processing speed can help our system to respond faster.”

Referee #3 -- Comment 5:

5. In the sub-sections of “Moving object detection and identification” and “RF signal detection” there exist some similarities in the description of experimental results. Please increase the readability and conciseness of the article.

Authors Response:

Thank you for pointing out this problem. We have rearranged these sub-sections, and also revised the manuscript and the supplementary material accordingly. All changes are highlighted in the revised documents.

Fig. R24. Experimental test of switching speed and actual sampling results at different send frame rates. (a, b) Experimental setup and results of FPGA response speed. Experimental setup of the response speed of the intelligent track system (as given in figure (c)). Results of the response speed of the intelligent track system, when the coding sequence is sent every three frames (as indicated in (d)) and once per frame (as indicated in (e)).

Reviewers' Comments:

Reviewer #1:

Remarks to the Author:

The authors have addressed all review comments from this reviewer in this revised manuscript with great details and solid work. The quality of this manuscript has been significantly improved and this work demonstrates good novelties and significant contributions. Therefore, an acceptance is recommended.

In addition, this reviewer appreciates the efforts that the authors have taken to consider the suggestions and make revisions.

Reviewer #2:

Remarks to the Author:

A remarkable work has been done by the authors to address all my questions.

In view of the consequent modifications made to the bibliography and to the scientific content of the paper, this demonstration now seems much better documented and the experimental evidence much more solid under more realistic conditions.

I recommend this paper for publication.

Reviewer #3:

Remarks to the Author:

The authors gave a positive response to the reviewers' comments. And some new analyses have been added in the revised manuscript. This manuscript is recommended for publication.

Response Letter to Reviewers

General comments from Referee #1:

The authors have addressed all review comments from this reviewer in this revised manuscript with great details and solid work. The quality of this manuscript has been significantly improved and this work demonstrates good novelties and significant contributions. Therefore, an acceptance is recommended.

In addition, this reviewer appreciates the efforts that the authors have taken to consider the suggestions and make revisions.

Authors Response:

Your insightful comments greatly helped us improve and strengthen this work. We are very pleased that you are happy with the changes we have made to the manuscript following your suggestions and questions. We would like to thank you once again for your help.

General comments from Referee #2:

A remarkable work has been done by the authors to address all my questions.

In view of the consequent modifications made to the bibliography and to the scientific content of the paper, this demonstration now seems much better documented and the experimental evidence much more solid under more realistic conditions.

I recommend this paper for publication.

Authors Response:

We sincerely thank you for the positive comments and the recommendation of our work. Your comments and questions are very important for us to improve the quality of this work. We would like to thank you once again for your help.

General comments from Referee #3:

The authors gave a positive response to the reviewers' comments. And some new analyses have been added in the revised manuscript. This manuscript is recommended for publication.

Authors Response:

Thank you so much for your professional comments, which helped us greatly improve the quality of this work. We are honored to have your approval of the revised article, and would like to thank you once again for your help.